# Generating Symbolic World Models
# via Test-time Scaling of Large Language Models

**Zhouliang Yu[1,2,\*],  Yuhuan Yuan[3,\*],  Tim Z. Xiao[4],  Fuxiang Frank Xia[5],  Jie Fu[6],
Ge Zhang[7],  Ge Lin[3,†],  Weiyang Liu[4,†]**

[1]The Chinese University of Hong Kong   [2]Hong Kong University of Science and Technology
[3]HKUST (Guangzhou)   [4]Max Planck Institute for Intelligent Systems, Tübingen
[5]Environmental Systems Research Institute   [6]Shanghai Artificial Intelligence Laboratory   [7]SEED, Bytedance

**Reviewed on OpenReview:** https://openreview.net/forum?id=zVo6PfBaOK

## Abstract

Solving complex planning problems requires Large Language Models (LLMs) to explicitly model the state transition to avoid rule violations, comply with constraints, and ensure optimality—a task hindered by the inherent ambiguity of natural language. To overcome such ambiguity, Planning Domain Definition Language (PDDL) is leveraged as a planning abstraction that enables precise and formal state descriptions. With PDDL, we can generate a symbolic world model where classic searching algorithms, such as A∗, can be seamlessly applied to find optimal plans. However, directly generating PDDL domains with current LLMs remains an open challenge due to the lack of PDDL training data. To address this challenge, we propose to scale up the test-time computation of LLMs to enhance their PDDL reasoning capabilities, thereby enabling the generation of high-quality PDDL domains. Specifically, we introduce a simple yet effective algorithm, which first employs a Best-of-N sampling approach to improve the quality of the initial solution and then refines the solution in a fine-grained manner with verbalized machine learning. Our method outperforms o1-mini by a considerable margin in the generation of PDDL domains, achieving over 50% success rate on two tasks (*i.e.*, generating PDDL domains from natural language description or PDDL problems). This is done without requiring additional training. By taking advantage of PDDL as state abstraction, our method is able to outperform current state-of-the-art methods on almost all competition-level planning tasks.

## 1 Introduction

Enabling large language models (LLMs) [22] to plan in complex scenarios like Barman, Floortile, and Termes remains an open problem [45]. While recent LLMs like OpenAI-o1-mini and OpenAI-o1-preview [22] excel at complex reasoning tasks, including coding [9] and mathematics [11], they still struggle with deductive reasoning and principled planning [10, 45, 43] that requires the consideration of optimality, constraints, and complex state transitions. This limitation persists in o1-preview even when using self-critique techniques and multiple answer re-sampling strategies (See Figure 2). A natural solution is translating the world abstraction from natural language into PDDL [27], which utilizes first-order logic (FOL) to explicitly describe states and relationships. Compared to natural language, the formal nature of PDDL simplifies verification and enables the precise specification of constraints and objectives, facilitating the seamless integration of off-the-shelf planning algorithms. However, it remains an open challenge to translate natural language descriptions into PDDL domains with accurate syntax and semantics. Current LLMs perform poorly in this translation task due to two key challenges: the scarcity of high-quality PDDL training data [61] and the complexity of

---

*Equal contribution. †Corresponding Authors.

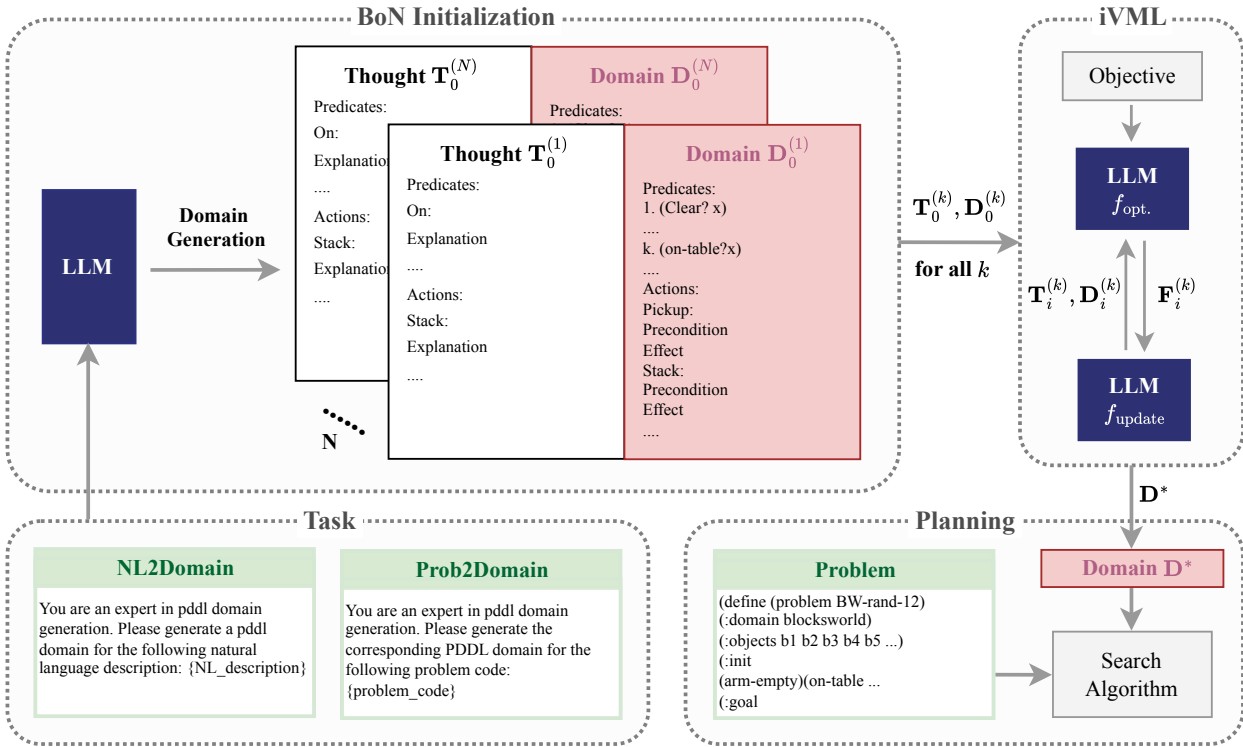

Figure 1: An overview of the proposed method. Our test-time compute scaling approach consists of two main steps: (1) Best-of-N Sampling for PDDL Initialization (see Section 3.2): We start by running a parallel sampling process to generate multiple chain-of-thought responses that are composed of the formalized PDDL-based world model representation $\mathbf{D}_0^{(k)}$ and the natural language thought $\mathbf{T}_0^{(k)}$. (2) Closed-loop Iteration with iVML (see Section 3.3): We use iVML to iteratively improve the solutions. iVML incorporates: (1) An optimizer LLM $f_{\text{opt}}$ that evaluates the solutions from the previous iteration, and (2) A learner LLM $f_{\text{update}}$ that learns from the feedback and updates the PDDL-based world model $\mathbf{D}_i$. Here, $N$ represents the total number of candidate solutions generated, $k$ is the index of the top candidates retained for further optimization (with $K$ indicating the total number of such candidates), and the index $i$ is used to denote the iteration step within the optimization procedure. The optimal PDDL-based world model will be used in the systematic search engine for planning.

maintaining logical consistency across predicates and actions [26]. Traditionally, the translation process has heavily relied on human expertise [17] or orale environment [37], making it difficult to automate and scale.

To address these challenges, our work leverages LLMs to generate PDDL-based symbolic world models for task planning without requiring model fine-tuning. We achieve this through a simple yet effective strategy to scale the test-time computation of LLMs. Specifically, we start by generating multiple PDDL domains from the input text query using Best-of-N (BoN) sampling [46]. This step aims to generate diverse candidate solutions for initializing the verbalized optimization process using high-temperature sampling. Then we refine the best candidates by optimizing the generated PDDL domains with the proposed instance verbalized machine learning (iVML) such that the generated PDDL domain can gradually fit the input query (*e.g.*, natural language descriptions or PDDL problems).

Our method is guided by the insight that effective test-time scaling can elicit stronger reasoning capabilities over formal languages like PDDL, thereby compensating for the scarcity of high-quality PDDL training data. We start by applying BoN sampling to explore the solution space and select the best initial solution, and then iVML aims to exploit the solution space around this initialization such that the solution gets gradually improved. Verbalized Machine Learning (VML; [50]) is a test-time in-context training framework where the main idea is to parameterize functions using text prompts. Such text-parameterised functions can be evaluated through LLMs. To solve a machine learning task, VML iteratively refines a learner LLM's text prompt based on feedback from an optimizer LLM, which takes into account the training data and learn-

ing objectives. The learner LLM's text prompt characterizes data patterns to perform inductive inference tasks such as regression and classification. In our paper, we apply VML to PDDL domain refinement and propose the iVML framework that reformulates VML for instance learning. Unlike the original tasks in [50], PDDL domain refinement is an instance learning task without training data to produce verbalized gradients. Therefore, rather than requiring training data, we use LLMs to verify the validity of PDDL domains and generate critiques that serve as verbalized gradients. Specifically, the learner LLM receives an initial PDDL domain and generates a refined version based on critiques from an optimizer LLM, iteratively eliminating logical inconsistencies and grammatical errors. However, the performance of iVML is highly dependent on the quality of initial solutions, as poor initialization can result in slow convergence or suboptimal solutions. While BoN sampling emphasizes exploration by independently generating diverse solutions, it does not leverage past predictions from LLMs, hence limiting its ability to exploit the solution space. iVML, in contrast, emphasizes exploitation by iteratively refining solutions based on the optimizer LLM's feedback. To achieve a good balance between exploration and exploitation, we propose to combine these complementary strategies for generating high-quality PDDL domains. This hybrid approach leverages the strengths of both methods by applying iVML to refine solutions initially generated through BoN sampling. Our contributions include:

- **Scalable PDDL domain generation**: We propose an effective test-time scaling approach for automatic and scalable PDDL domain generation without additional model training. Using Qwen2.5-Coder-7B as the base model, our approach achieves state-of-the-art performance with an 85.2% success rate on the NL2Domain task and 71.4% on Prob2Domain, substantially surpassing o1-mini's performance (41.7% and 33.7%, respectively).

- **Application of VML to instance learning**. We introduce the iVML framework to adapt VML to instance learning, where no training data is available. We use LLMs to check the validity of PDDL domains and generate textual critiques as gradients to iteratively update PDDL domains.

- **Efficient test-time compute scaling**: We enhance VML with BoN sampling initialization, effectively balancing exploration and exploitation to achieve faster convergence and obtain better solutions.

- **Robust planning through PDDL abstraction**: We demonstrate that PDDL-based formal abstraction enables more robust planning compared to direct LLM-based approaches. Our method successfully handles complex domains such as Barman and Termes, where existing LLM-based planners fail.

## 2 Related Work

### 2.1 LLMs for Task Planning

Recent advances in large language models (LLMs) (*e.g.*, GPT-4 [1]) have shown promise in handling common reasoning tasks, such as GSM8K [11] and HumanEval [8]. The gain of reasoning capabilities of LLMs can be attributed to (but not limited to) several factors: (1) Extensive training on reasoning datasets: [54] fine-tunes LLMs with distilled chain-of-thought datasets to improve plausible reasoning, and recent models have scaled this approach using larger and more diverse datasets, which has enhanced performance on tasks such as mathematical reasoning, coding, and logical reasoning. However, such an approach raises concerns about whether the observed gains are attributable to data contamination [39, 28]. (2) Self-improvement during inference time: Some approaches incorporate verifiers that provide synthetic feedback during inference, using self-reflection [36], process reward models [57], or simple sparse objective reward [59] to guide improvement. However, [39] shows that imperfect verifiers increase false positive samples during scaling up reasoning at inference time. Recent advanced reasoning-focused LLMs (*e.g.*, o1-mini and o1-preview) have adopted both strategies to train extensively on reasoning datasets [22] and leverage the long hidden-chain of thought [48] during inference, making them essentially different from vanilla LLMs (*e.g.*, GPT-4). However, both GPT-4 and o1-preview exhibit limitations in tackling complex planning tasks. As demonstrated by [44], neither GPT-4 nor o1-preview succeeded in solving any instances of very complex planning problems such as Termes, Barman, and Floortile. While o1-preview demonstrated marginally higher accuracy on relatively simpler planning tasks (*e.g.*, BlockWorld and Tyreworld), they are observed making frequent hallucination by rule violation, inabilities to generate feasible plans, misinterpretation of goal state, etc.

Figure 2: OpenAI-o1-preview plans for Termes: o1-preview frequently exhibits hallucination during the planning process. Specifically, in steps three and four, the LLM violates predefined rules when selecting and leveraging actions. Additionally, step four hallucinates the achievement of the goal, leading to incorrect or unrealistic outcomes. Even when using o1-preview itself to evaluate the hallucinated plan, it incorrectly identifies the plan as valid.

Although vanilla LLMs can approximate state transitions [18], they do so by probabilistically predicting subsequent tokens based on patterns learned from vast datasets rather than through logical deduction or structured inference [24]. According to [43], o1-preview similarly relies on approximate reasoning during planning, executing fuzzy pattern matching that stems from its reinforcement learning training paradigm. This approach, while effective for certain reasoning tasks, fundamentally differs from structured logical inference [10]. Inspired by traditional model-based reinforcement learning approaches [2], which solve decision-making problems by predicting discrete world models for heuristic search, our method significantly enhances LLMs' planning capabilities through a two-stage process. First, we predict a symbolic Planning Domain Definition Language (PDDL)-based world model using an instance verbalized machine learning approach. This stage transforms the problem into a structured, symbolic representation, enabling more logical and systematic reasoning. Second, we leverage heuristic search methods, such as A∗, to efficiently find optimal solutions within this structured framework.

| Methods | Synthesis Objective | Benchmarking | Technical Method |
|---|---|---|---|
| [17] | Each action separately within PDDL domains | 3 domains: Household, Logistics, Tyreworld | Human experts |
| [61] | Whole PDDL problems (initial state and goal) | 2 domains: Gripper, Blockworld | Finetuned on a large dataset |
| [26] | Whole PDDL domains | 10 domains: Gripper, Blockworlds, Termes, etc | Interact with oracles (*i.e.*, Require ground-truth environment) |
| Ours | Whole PDDL domains | 283 IPC domains (NL2Domain) 332 IPC domains (Prob2Domain) | Test-time scaling without model training & oracle interaction |

Table 1: Comparison between current PDDL synthesis methods and ours.

## 2.2 World Model Generation

However, automatically generating scalable PDDL-based world models by LLMs is still challenging. Current LLMs rely heavily on either human-in-the-loop, extensive training data, or oracle environment to plausibly be superior in generating PDDL on limited scenarios. For example, [17] leverages multiple human experts to refine the logical error in individual PDDL action expressions generated by LLMs. [61] collect more than 132,027 SFT data to train LLM on limited simple planning scenarios, (*e.g.*, BlockWorld and Grippers). Meanwhile, [30, 26, 38] employ iterative refinement processes for generated PDDL domains. These approaches access oracle environments—consisting of ground-truth PDDL problems and PDDL domains—and validate the generated PDDL domain by the feedback that compares the resulting plans against ground-truth plans generated by the oracles. By contrast, our approach generates PDDL domains in an oracle-agnostic manner, focusing on scalability to scenarios where rule-based ground-truth signals are unavailable. To compare with other PDDL-related works, our method eliminates the need for training data and does not rely on an oracle to improve the generated PDDL domain (See Table 1).

Several methods have also explored using LLMs to generate world model representations other than PDDL. For example, GIF-MCTS [12] and World-coder [40] translate natural language descriptions of world models into Gym-like Python code [6], and use pre-collected trajectories to validate and provide feedback for wrong state-transition predictions iteratively. Our work, however, focuses on more general planning scenarios without pre-collected validation datasets to provide critique feedback.

## 2.3 Adaptation of LLMs

In the absence of pre-collected validation datasets, the adaptation of LLMs for the PDDL-based world model presents unique challenges. While parameter-efficient finetuning methods (*e.g.*, [21, 33, 25, 13]) enable effective adaptation of LLMs to downstream tasks, they still require high-quality training data. If there is insufficient training data, in-context learning [7, 47, 14] offers an alternative adaptation approach. Prompt optimization techniques [60, 31, 52] further enhance adaptation performance by deriving improved instruction prompts from limited training data. Recently, [50] proposes verbalized machine learning (VML) to update model-characterizing text prompts with LLMs. Unlike conventional machine learning settings, functions in VML are parameterized within the language space, and the "model" in the text prompt is iteratively refined through the in-context learning mechanisms of LLMs. [31, 55] introduce the concept of textual gradients as criteria for updating text prompts. Although finetuning methods can achieve effective adaptation, they risk causing catastrophic forgetting in pretrained LLMs, potentially compromising their general instruction-following capabilities. Therefore, test-time adaptation of LLMs to downstream tasks has emerged as a practical solution. In our paper, we introduce a simple test-time adaptation method for generating PDDL-based world models, which scales efficiently with test-time compute.

# 3 The Proposed Test-time Compute Scaling Approach

Natural language task planning faces significant challenges in state estimation, constraint-based plan generation, and plan validation. One approach to address this issue is to generate explicit, unambiguous world models (*e.g.*, PDDL) before conducting planning accordingly. Our proposed methods aim to explore the key question: *How can we effectively generate symbolic world model via LLMs?*

Generating symbolic world models requires not only natural language understanding but also sophisticated deductive reasoning to maintain logical consistency across all model components. Without such formal modeling sophistication, models risk generating inconsistent state transitions and producing suboptimal plans. To address this challenge, we enhance LLMs' reasoning capabilities through an instance verbalized machine learning algorithm (see Section 3.3), initializing it with better candidates generated by the best-of-N sampling. See Figure 1 for the overview of our method.

## 3.1 PDDL-based World Model Representation

Classical planning problems are inherently complex. Even determining plan satisfiability [34]—whether any solution exists for a given planning problem—is NP-hard. Planning problems that involve optimization under constraints pose even greater challenges for natural language planners. Our approach, which utilizes the PDDL-based representation, offers several advantages: (1) PDDL employs a logical system to express atoms and predicates derived from STRIPS [16] (*i.e.*, Stanford Research Institute Problem Solver), and therefore it provides a formal and unambiguous syntax for representing world models. PDDL-based world modeling employs explicit representations that not only enrich the description of actions and states but also ensure precise and straightforward validation, thereby eliminating ambiguity. (2) Natural language plan prediction often reduces to an n-gram task of ungrounded token generation. We provide the description of the Termes problem in both natural language (see Figure 2) and PDDL (see the text box below) to illustrate the difference. By adopting PDDL as representation, we transform this task into explicit classical planning. This formulation enables the use of search algorithms such as A∗. With the help of the PDDL representation, a planning graph (*e.g.*, Figure 6) can be used to provide improved heuristic estimates when searching through the state space.

## 3.2 Best-of-N Sampling for PDDL Initialization

Our test-time scaling approach adopts a two-stage optimization process. The first stage is to search for good initial solutions via best-of-N sampling, and the second stage is to iteratively refine the initial solutions via an instance verbalized machine learning algorithm (See Section 3.3). For each problem, the LLM generates many candidate solutions in parallel and retains the $K$ samples with the highest log-likelihoods. This process involves three main steps: candidate generation, scoring, and selection. During sampling, a temperature parameter $\tau$ is used to scale the logits (*i.e.*, the log-likelihood of raw score associated with a certain token), and in this case we can represent the probability of token $w_t^{(i)}$ at position $t$ as:

$$p_t(w_t^{(i)}) = \frac{\exp\left(\frac{\text{logit}(w_t^{(i)})}{\tau}\right)}{\sum_{v \in \mathcal{V}} \exp\left(\frac{\text{logit}(v)}{\tau}\right)}, \tag{1}$$

where the temperature $\tau$ adjusts the sampling distribution. Higher temperature (*e.g.* $\tau \geq 0.7$) increases randomness and diversity in the generated candidates by flattening the probability distribution, giving relatively more weight to lower-probability tokens, in contrast, a lower temperature (*e.g.* $\tau \leq 0.3$) sharpens the distribution, favoring high-probability tokens and producing more deterministic outputs. In the scoring phase, each candidate $c_i$ is assigned a score $S_i$ based on the sum of the log-likelihoods of its generated tokens:

$$S_i = \sum_{t=1}^{L_i} \log p_t\left(w_t^{(i)}\right), \quad \forall i \in \{1, 2, \ldots, N\} \tag{2}$$

where: $L_i$ is the length of candidate $c_i$. In the selection phase, the top $K$ candidates with the highest scores are chosen as the initialization points to be optimized by iVML.

---

**Termes in Planning Domain Definition Language**

**Domain**:

```
(define (domain termes)
    (:requirements :typing :negative-preconditions)
    (:types
        numb - object
        position - object)
    (:predicates
        (height ?p - position ?h - numb)
        (at ?p - position)
        (has-block)
        (SUCC ?n1 - numb ?n2 - numb)
        (NEIGHBOR ?p1 - position ?p2 - position)
        (IS-DEPOT ?p - position))
    (:action move
        :parameters (?from - position ?to - position ?h - numb)
        :precondition (and (at ?from)(NEIGHBOR ?from ?to)(height ?from ?h)(height ?to ?h))
        :effect (and (not (at ?from))(at ?to) ) )
    (:action move-up
        :parameters (?from - position ?hfrom - numb ?to - position ?hto - numb)
        :precondition (and(at ?from)(NEIGHBOR ?from ?to)(height ?from ?hfrom)(height
        ?to ?hto)(SUCC ?hto ?hfrom))
        :effect (and(not (at ?from))(at ?to)))

    (:action move-down ...
    (:action place-block ...
    (:action remove-block ...
    (:action create-block ...
    (:action destroy-block ...
)
```

**Problem**:

```
(define (problem termes-00038-0036-4x3x3-random_towers_4x3_3_1_3)
(:domain termes)
; termes-00038-0036-4x3x3-random_towers_4x3_3_1_3
; Initial state:
;  0   0   R0D  0
;  0   0   0    0
;  0   0   0    0
; Goal state:
;  0   0   0    0
;  0   0   0    0
;  0   3   0    0
; Maximal height: 3
(:objects
    n0 - numb......
    pos-0-0 - position......
)
(:init
    (height pos-0-0 n0)......
    (at pos-2-0)
    (SUCC n1 n0)......
    (NEIGHBOR pos-0-0 pos-1-0)......
    (IS-DEPOT pos-2-0)
)
(:goal
(and (height pos-0-0 n0) ...... (not (has-block)))))
```

### 3.3  iVML: Instance Verbalized Machine learning

With the BoN sampling to select the candidates as the initial solution, we introduce instance verbalized machine learning to refine both the generated PDDL domain and the natural language chain of thought (CoT). iVML is an adaptation of VML [50] to the instance optimization setting, where the goal is to optimize and refine a single instance (*i.e.*, PDDL domains in this paper). In iVML, the functions are parameterized using natural language rather than numerical values in conventional machine learning. Viewing an LLM as the inference engine, we can evaluate such a natural language parameterized function (*i.e.*, via $f_{opt}(\cdot)$), and optimize its model parameters in the natural language space (*i.e.*, via $f_{update}(\cdot)$). In the context of iVML, the optimization objective is to generate a PDDL domain $D^*$ that correctly aligns with the input prompt description $\mathcal{G}$, as formulated below

$$\mathbf{D}^* = \arg\min_{\mathbf{D}} \ \mathcal{L}(\mathcal{G}, \mathbf{D}), \tag{3}$$

where $\mathcal{L}(\cdot)$ is a loss function defining the closeness between $\mathcal{G}$ and $\mathbf{D}$. Here, $\mathcal{G}$ is always within the language spaces for all the tasks we considered, for example, in NL2Domain task, $\mathcal{G}$ corresponds to a natural language description of the planning domain, for Prob2Domain, $\mathcal{G}$ represents a PDDL problem instance. Solving Equation (3) is difficult as both $\mathcal{G}$ and $\mathbf{D}$ are text, and $\mathcal{L}(\cdot)$ is hard to define unless abstractly using natural language. Using the iVML framework, we can approximately solve Equation (3) with an iterative algorithm that alternates between two natural language parameterized functions at the iteration $i$:

$$\mathbf{F}_i = f_{\text{opt}}(\mathcal{L}, \mathcal{G}, \mathbf{T}_{i-1}, \mathbf{D}_{i-1}), \quad \mathbf{T}_i, \mathbf{D}_i = f_{\text{update}}(\mathbf{F}_i, \mathbf{T}_{i-1}, \mathbf{D}_{i-1}), \tag{4}$$

where $\mathbf{T}_{i-1}$ and $\mathbf{D}_{i-1}$ correspond to the current thoughts and the current PDDL domain, $\mathbf{F}_i$ is the feedback from the optimizer function $f_{\text{opt}}(\cdot)$, $\mathbf{T}_i$ and $\mathbf{D}_i$ are the updated thoughts and PDDL domain output from the update function $f_{\text{update}}(\cdot)$. Here, $\mathbf{T}_i$ is generated by applying CoT sampling at each step of the PDDL domain generation process, it comprises all non-PDDL components in the outputs of $f_{\text{update}}(\cdot)$, excluding the final structured PDDL domain itself. Both $f_{\text{opt}}(\cdot)$ and $f_{\text{update}}(\cdot)$ are parameterized by natural language. The initial PDDL domain $\mathbf{D}_0$ is initialized from the best-of-N sampling. These two functions are evaluated through separate LLMs calls. We show the prompt templates for $f_{\text{opt}}(\cdot)$ and $f_{\text{update}}(\cdot)$ below:

---

**Prompt template for $f_{\text{opt}}(\cdot)$**

You will be provided a natural language description of a planning domain, and its corresponding PDDL domain code with intermediate thoughts explaining each predicate and action. Your task is to generate critical feedback on the PDDL domain code based on the natural language description. You should evaluate the grammar and logic of the PDDL domain codes, and the logic error in the intermediate thoughts.
PDDL synthesis problem: $\{\mathcal{G}\}$
natural language chain of thoughts: $\{\mathbf{T}_{i-1}\}$
Generated PDDL domain: $\{\mathbf{D}_{i-1}\}$

---

**Prompt template for $f_{\text{update}}(\cdot)$**

You will be provided a PDDL domain code and critical feedback on the PDDL domain code based on the natural language description. Your task is to generate a new PDDL domain code that is more consistent with the natural language description.
PDDL synthesis problem: $\{\mathcal{G}\}$
Natural language chain of thoughts at the previous turn: $\{\mathbf{T}_{i-1}\}$
Generated PDDL domain at the previous turn: $\{\mathbf{D}_{i-1}\}$
The error of the PDDL domain $\{\mathbf{F}_{i-1}\}$

---

**iVML with BoN initialization effectively balances exploration and exploitation.** BoN employs a broad exploration strategy, maintaining diverse candidate solutions to probe distinct regions of the combinatorial search space. While this approach mitigates initialization bias, it suffers from diminishing returns: beyond a critical sample size, the probability of discovering novel valid solutions decays due to redundant model generations. In contrast, iVML performs verbalized in-context exploitation by iteratively refining BoN-selected candidates based on objectives defined in natural language. This closed-loop process enables precise error correction (*e.g.*, resolving precondition conflicts in PDDL actions) but remains susceptible to local minima—a fundamental challenge in non-convex optimization [35]. Our approach combines BoN and iVML to achieve an effective balance between exploration and exploitation, addressing the limitations of each method alone through a two-phase optimization framework: (1) BoN initialization that generates multiple diverse and high-quality initializations $\{\mathbf{D}_0^{(i)}\}_{i=1}^{k}$, and (2) iVML refinement that optimizes these initial solutions with verbalized machine learning.

## 4 Experiments and Results

Section 4.2 examines state-of-the-art LLMs' capabilities on PDDL domain synthesis and evaluates whether test-time scaling approaches enhance PDDL generation quality. Section 4.3 analyzes the convergence patterns of BoN and iVML, highlighting the specific improvements iVML offers over BoN. Section 4.4 investigates iVML's sensitivity to initialization conditions and whether diverse candidate sets can boost iVML performance. Section 4.5 explores whether iVML's advantages in PDDL domain synthesis extend to other symbolic code generation tasks within the PDDL language (*e.g.*, PDDL problem synthesis), demonstrating its broader applicability. Section 4.6 assesses our planning approach (generating PDDL domains via iVML followed by heuristic search-based planning) against LLM-based planners on complex planning problems.

| Model | Params | NL2Domain (%) | Problem2Domain (%) | Avg. (%) |
|---|---|---|---|---|
| *Open-Source Models* | | | | |
| Qwen2.5-Instruct | 0.5B | 0.0 | 0.0 | 0.0 |
| Qwen2.5-Instruct | 1.5B | 0.0 | 0.0 | 0.0 |
| Qwen2.5-Instruct | 3B | 2.1 | 1.5 | 1.8 |
| Qwen2.5-Instruct | 7B | 5.7 | 11.7 | 8.7 |
| Qwen2.5-Instruct | 14B | 21.6 | 25.3 | 23.5 |
| Qwen2.5-Instruct | 32B | 24.0 | 31.6 | 27.8 |
| Qwen2.5-Instruct | 72B | 38.5 | 32.8 | 35.7 |
| Qwen2.5-Coder | 1.5B | 0.0 | 0.0 | 0.0 |
| Qwen2.5-Coder | 7B | 21.9 | 18.4 | 20.2 |
| Llama3.1-Instruct | 8B | 0.0 | 0.0 | 0.0 |
| Llama3.1-Instruct | 70B | 1.1 | 0.0 | 0.6 |
| Yi-1.5-Chat | 6B | 0.4 | 1.8 | 1.1 |
| Yi-1.5-Chat | 9B | 6.7 | 9.3 | 8.0 |
| Yi-1.5-Chat | 34B | 12.0 | 8.7 | 10.4 |
| Yi-Coder | 1.5B | 0.0 | 0.0 | 0.0 |
| Yi-Coder | 9B | 9.9 | 14.5 | 12.2 |
| *Closed-Source Models* | | | | |
| GPT-4o | - | 5.3 | 50.0 | 27.7 |
| o1-mini | - | 41.7 | 33.7 | 37.7 |
| o1-preview | - | 55.8 | 52.4 | 54.1 |
| *Our Methods* | | | | |
| BoN-8-Qwen2.5-Instrcut | 0.5B | 0.0 (+0.0) | 0.0 (+0.0) | 0.0 |
| BoN-8-Qwen2.5-Instrcut | 1.5B | 2.1 (+2.1) | 0.3 (+0.3) | 1.2 |
| BoN-8-Qwen2.5-Instrcut | 3B | 11.7 (+9.5) | 1.2 (+0.3) | 6.5 |
| BoN-8-Qwen2.5-Instrcut | 7B | 9.2 (+3.5) | 34.6 (+22.9) | 21.9 |
| BoN-8-Qwen2.5-Instrcut | 14B | 51.6 (+30.0) | 62.0 (+36.7) | 56.8 |
| BoN-8-Qwen2.5-Instrcut | 32B | 66.8 (+46.7) | 71.1 (+39.5) | 70.9 |
| BoN-8-Qwen2.5-Instruct | 72B | 60.8 (+22.3) | 73.8 (+41.0) | 67.3 |
| BoN-8-Qwen2.5-Coder | 7B | 73.1 (+51.2) | 63.3 (+44.9) | 68.2 |
| BoN-8-Yi-1.5-Chat | 9B | 46.6 (+39.9) | 39.8 (+30.5) | 43.2 |
| BoN-8-Llama3.1-Instruct | 8B | 0.7 (+0.7) | 0.0 (+0.0) | 0.4 |
| iVML-5-BoN-8-Qwen2.5-Instruct | 0.5B | 0.0 (+0.0) | 0.0 (+0.0) | 0.0 |
| iVML-5-BoN-8-Qwen2.5-Instruct | 1.5B | 2.8 (+2.8) | 0.3 (+0.3) | 1.6 |
| iVML-5-BoN-8-Qwen2.5-Instruct | 3B | 18.7 (+16.6) | 1.8 (+0.3) | 10.3 |
| iVML-5-BoN-8-Qwen2.5-Instruct | 7B | 21.9 (+16.2) | 49.1 (+37.3) | 35.5 |
| iVML-5-BoN-8-Qwen2.5-Instruct | 14B | 77.0 (+55.4) | 80.4 (+55.1) | 78.7 |
| iVML-5-BoN-8-Qwen2.5-Instruct | 32B | 86.2 (+62.2) | 90.9 (+59.3) | 88.6 |
| iVML-5-BoN-8-Qwen2.5-Instruct | 72B | 78.4 (+39.9) | 86.4 (+53.6) | 82.4 |
| iVML-5-BoN-8-Qwen2.5-Coder | 7B | 85.2 (+63.3) | 71.4 (+53.0) | 78.3 |
| iVML-5-BoN-8-Yi-1.5-Chat | 9B | **93.9 (+84.0)** | **93.0 (+78.5)** | **93.5** |
| iVML-5-BoN-4-Llama3.1-Instruct | 8B | 2.8 (+2.8) | 0 (+0.0) | 1.4 |

Table 2: A comparison of performance in PDDL domain synthesis between the baseline models (including both open-source and closed-source models) and our methods. BoN-8 refers to BoN sampling with 8 candidates, while iVML-5-BoN-8 denotes five iterations of iVML training initialized with BoN-8. In this experiment, the metric refers to the percentages of the generated PDDL domains that pass VAL validation without errors.

## 4.1 Experiment Setup

**Evaluation tasks and datasets**. We evaluate several methods on the International Planning Competition benchmark[1], which encompasses diverse complex planning domains and problems. Our evaluation focuses on two key PDDL domain synthesis tasks, including (1) NL2Domain, which aims to convert natural language descriptions to PDDL Domains; and (2) Prob2Domai, which aims to derive necessary PDDL domains from PDDL problems. The evaluation metric used here is the success rate of the generated PDDL domain passing the PDDL validation system [20].

**Large language model settings**. The backbone LLMs in our experiment include Qwen2.5-Instruct (0.5B-72B parameters) [51], LLaMA3.1-Instruct (8B and 70B parameters) [15], and Yi-1.5-Chat (6B, 9B, and 34B

---

[1]https://github.com/potassco/pddl-instances

parameters) [53]. We also incorporate specialized code-oriented LLMs, specifically Qwen2.5-Coder and Yi-1.5-Coder. In addition to open-source LLMs, we benchmark against OpenAI's proprietary models, including GPT-4o, o1-mini, and o1-preview. We test our proposed test-time scaling methods on Qwen models in a zero-shot setting without model finetuning.

**Chain of thought prompting**. All baselines here utilize chain-of-thought (CoT) prompting by default. Current LLMs have been extensively trained on datasets that include step-by-step reasoning besides final answers [49]. This enables the models to generate better reasoning traces during inference. The detailed CoT prompt template for our method is provided in Appendix D.

**Sampling hyperparameters**. To generate diverse PDDL domain synthesis paths, we use temperature sampling for both the BoN and iVML algorithms, and the temperature equals 0.7.

**Evaluation Metrics**. We clarify the evaluation metrics for the experiments as follows:

- For PDDL domain synthesis experiments (*i.e.*, from Section 4.2 to Section 4.4), we adopt the VAL [20] validation tool to examine the syntax and semantic correctness of the PDDL domains. This includes checking for issues such as invalid predicate parameter types, incorrect predicate names/formats, or improper predicate usage. For methods generating PDDL domains, we measure the percentage of domains that pass VAL validation without errors.

- For PDDL problem synthesis experiment (*i.e.*, Section 4.5), we use the percentage of correct generated problems (*i.e.*, the generated symbolic goals matches with the ground-truth goals) as metric.

- For the experiments in Section 4.6, we evaluate planning accuracy, defined as the percentage of synthesized domains that produce plans identical to the ground-truth reference plans.

## 4.2 Main Results in PDDL Domain Synthesis

**Current LLMs perform poorly in PDDL domain synthesis.** Despite advances in code and math reasoning, LLMs exhibit fundamental limitations in PDDL-based formal synthesis. For instance, in Table 2, Qwen2.5-Instruct (72B) achieves only 38.5% and 32.8% accuracy in NL2Domain and Prob2Domain tasks, respectively. The results suggest that existing LLMs still fall short in symbolic reasoning tasks.

**Search-augmented reasoning enhances formal synthesis.** In Table 2, among the closed-source models, o1-preview emerges as the top performer with an average accuracy of 54.1%, outperforming other models in both NL2Domain and Problem2Domain tasks. The o1-series models, which integrate search-based reasoning during inference [32, 56], demonstrate significant improvements over standard instruction LLMs. For example, GPT-4o achieves only an average accuracy of 27.7%.

**Code-oriented models outperform general-purpose models.** In Table 2, we observe that code-specialized models (*e.g.*, Qwen2.5-Coder and Yi-1.5-Coder) demonstrate superior performance than their general-purpose counterparts. For example, Qwen2.5-Coder (7B) outperforms Qwen-Instruct (7B) by 16.2% NL2Domain (*i.e.*, 21.9% compared to 5.7%). We hypothesize that the improvements stem from: (1) Implicit formalization training: code datasets teach type systems and predicate logic, (2) Syntax-sensitive decoding: token-wise likelihood aligns with PDDL's Lisp-like structure, and (3) Autoformalization priors: the code datasets that interleave natural language comments with pieces of code are high-quality datasets for CoT reasoning and autoformalization.

**Test-time scaling is helpful for LLM at almost all scales.** The BoN sampling method demonstrates universal effectiveness across the Qwen model family (*e.g.*, 1.5B to 72B parameters), significantly improving PDDL domain synthesis accuracy. For example in Table 2, BoN sampling with 8 candidates (BoN-8) improves Qwen2.5-Instruct (14B) from 21.6% to 51.6% on NL2Domain. Gains persist through Qwen2.5-Instruct (72B) with 22.3% improvement on NL2Domain. Test-time compute scaling requires no further training or architectural changes, making it a computationally efficient addition to scaling up LLM's parameter numbers.

**iVML can provide a robust and consistent improvement.** iVML delivers robust performance gains over BoN across multiple model scales, demonstrating the power of iterative self-improvement in PDDL domain synthesis. As illustrated in Table 2, five iterations of iVML training with BoN-8 initialization (iVML-5-BoN-8) enables Qwen2.5-Instruct (32B) to achieve 86.2% on NL2Domain, outperforming base BoN-8 with

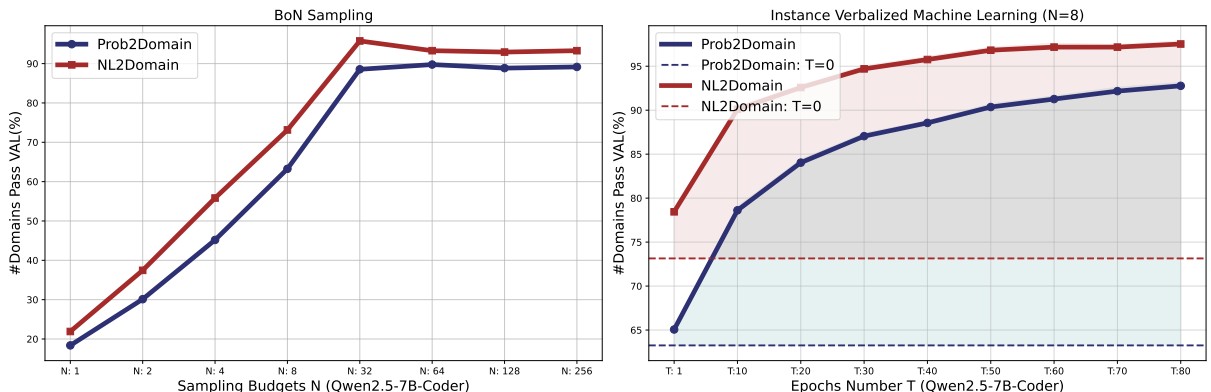

Figure 3: Left: The performance trend of BoN with increasing sampling numbers. Right: The performance trend of iVML with increasing training epochs (Setting the BoN sampling number N = 8).

19.4% improvement. The results position iVML as a scalable and efficient framework for enhancing LLM performance in formal synthesis tasks.

### 4.3 Convergence Comparison between BoN and iVML

**Experiment settings**. This section presents a comparative analysis of the convergence behavior of BoN and iVML in synthesis tasks of PDDL domains. The computational efficiency and synthesis success rates of these methods depend on two parameters: the sampling budget for BoN $N$, and the number of training epochs for iVML $T$. Through controlled experiments, we examine how parametric variations affect synthesis effectiveness and identify the conditions under which their performance converges. We use Qwen2.5-Coder (7B) as the backbone LLM. The initialization of iVML is based on BoN-8 if not otherwise specified.

**Convergence of BoN and iVML**. On the left of Figure 3, we observe two distinct phases in the performance of BoN sampling as the sampling budget $N$ increases. Phase 1 ($N \leq 32$): Both Prob2Domain and NL2Domain demonstrate a steep increase in the percentage of domains passing validation as the sampling budget increases, demonstrating the efficiency of exploring the candidate sampling to improve accuracy. Phase 2 ($N > 32$): Performance reaches saturation with observable degradation trends. At higher sampling budgets (32 and beyond), Prob2Domain and NL2Domain converge to around 90%. On the right of Figure 3, iVML exhibits a monotonic performance improvement up to $T = 80$, significantly surpassing the saturated success rate achieved by BoN. For example, at $N = 256$, BoN becomes saturated, with NL2Domain's accuracy falling below 95% and Prob2Domain not exceeding 90%. In contrast, iVML enhances the accuracy to over 95% for NL2Domain and surpasses 90% for Prob2Domain at $T = 80$, showing its superiority over BoN.

**Case study**. The qualitative case study is given in Table 3, where we show BoN often fails to generate the correct code. For example, in TyreWorld, despite explicitly stating the precondition that the container is open, BoN still generates the invalid predicate "closed ?container". Unlike BoN's brute-force sampling approach, iVML leverages an in-context self-refinement mechanism to (1) identify constraint violations (*e.g.*, illegal block stacking or invalid preconditions); (2) generate counterfactual natural language feedback to guide revisions; (3) strengthen domain-specific reasoning priors through iterative updates. Consequently, iVML can fix the BoN error and generate the corrected predicate "not (closed?container)"

**Analysis**. In Figure 3, we observe BoN's early saturation and performance degradation. This observation aligns with inference scaling flaws [39], where the optimal number of resampling steps for code synthesis tasks (*e.g.*, HumanEval [3] and MBPP [8]) is always finite and low. This phenomenon arises from the lack of in-context learning with feedback from a verifier. However, iVML uses a self-critique mechanism combined with in-context learning to create a synthetic curriculum, enabling the model to learn to avoid error patterns in PDDL synthesis progressively. In contrast, BoN struggles with such dependencies due to its reliance on static sampling and the lack of iterative refinement.

| Natural Language | BoN@8 | BoN@8 + iVML@5 |
|---|---|---|
| BlockWorld (stack): The arm can stack a block on top of another if it holds the top block and the bottom block is clear. Post-action, the arm is empty, the top block rests on the bottom block, and the bottom block is no longer clear. | (:action stack :parameters (?top - block ?bottom - block) :precondition (and (holding ?top) (clear ?bottom)) :effect (and (not (holding ?top)) (on ?top ?bottom) (clear ?bottom)) | Incorrectly clear the bottom block, which should not happen when stacking a block on the bottom block. (:action stack :parameters (?top - block ?bottom - block) :precondition (and (holding ?top) (clear ?bottom)) :effect (and (arm-empty) (not (holding ?top)) (not (clear ?bottom)) (on ?top ?bottom))) |
| TyreWorld(fetch): The precondition for this action is that the object is inside the container and the container is open. The effect of this action is that the object is held by the agent and not inside the container. | (:action fetch :parameters (?obj - object ?container - container) :precondition (and (in ?obj ?container) (closed ?container)) :effect (and (in ?obj ?container) (not (in ?obj ?container))) | The logic for the fetch action is incorrect. Precondition: Incorrectly checks if the object is inside the container and the container is open. Effect: Incorrect logic. The effect should be "(not (in ?obj ?container))" instead of "(in ?obj ?container) (not (in ?obj ?container))". (:action fetch :parameters (?obj - object ?container - container) :precondition (and (in ?obj ?container) (not (closed ?container))) :effect (not (in ?obj ?container)) |
| TSP(return-against): This action also completes the tour by allowing the agent to return to the starting city against the direction of a connection. The precondition is similar to the "return-along" action, except the city is not previously visited. The effect is that the agent returns to the starting point and the tour is marked as complete. | (:action return-against :parameters (?current-city - city ?start-city - city) :precondition (and (in ?current-city) (connected ?start-city ?current-city)(not-complete)) :effect (and (in ?start-city) (complete))) | The precondition for this action should have "(connected ?current-city ?start-city)" instead of "(connected ?start-city ?current-city)" to correctly reflect the travel direction. (:action return-against :parameters (?current-city - city ?start-city - city) :precondition (and (in ?current-city) (connected ?current-city ?start-city) (not-complete)) :effect (and (in ?start-city) (complete))) |
| Termes(remove-block): Create a block at the depot. The robot will have the block. | (:action remove-block :parameters (fromPos toPos) :precondition (and (height fromPos ?h) (SUCC ?h ?h-down) (height toPos ?h-down) (at fromPos) (not (has-block))) :effect (and (has-block) (height toPos ?h))) | The effect of the "remove-block" action does not correctly update the height of the block's position. It should be "(height toPos ?h-down)", but it is currently "(?h-down)". (:action remove-block :parameters (fromPos toPos) :precondition (and (height fromPos ?h) (SUCC ?h ?h-down) (height toPos ?h-down) (at fromPos) (not (has-block))) :effect (and (has-block) (height toPos ?h-down))) |

Table 3: The comparison highlights the differences between Best-of-N sampling (BoN) and iVML in synthesizing action-level PDDL code. The red text marks where BoN@8 produces logically incorrect code, while the blue text shows how iVML detects these inaccuracies and applies the necessary corrections.

## 4.4 Ablation Study of Initialization Strategies

**Experiment settings**. This study aims to compare two initialization strategies for iVML: 1. Single-pass sampling by requesting the LLMs to generate only once as the solution initialization for the iVML process, and 2. BoN sampling by requesting the LLMs to generate diverse candidate solutions for initializing the optimization process using high-temperature sampling.

**BoN vs. Single-pass sampling**. We compare BoN sampling with Single-pass sampling, which BoN sampling, as an initialization strategy, provides iVML with the dual advantages of accelerated convergence and improved solution quality. For example, in Figure 4, with BoN-8 initialization in NL2Domain, Deepseek-Coder saturates earlier than its single-pass counterpart at $T = 16$, while achieving a higher accuracy percentage (approximately 95% compared to fewer than 60% in single-pass). Unlike single-pass sampling, which is analogous to random initialization in traditional optimization, *e.g.*, stochastic gradient descent, BoN generates a diverse set of initial candidate solutions. The solution diversity can effectively improve iVML with expanded exploration. By covering a broader range of the solution space, BoN helps to avoid early convergence to suboptimal solutions. This means that the algorithm is less likely to get stuck in local minima, which are common challenges in complex optimization problems [4]. By selecting high-quality initial solution candidates, BoN guides the optimization process of iVML toward better optimality.

**Performance of weaker LLMs**. From Figure 4, we observe that iVML with BoN-4 initialization can improve LLaMa's performance in the NL2Domain task, which yields around 10 correct domains (around 4%) compared to zero in single-pass mode. However, the performance is much worse than Qwen2.5-Coder (7B) and Deepseek-Coder-Instruct-v1.5 (7B). We attribute this performance gap to LLaMa's limited exposure

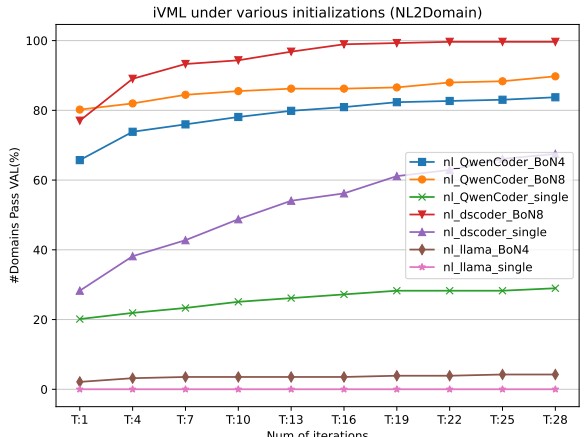

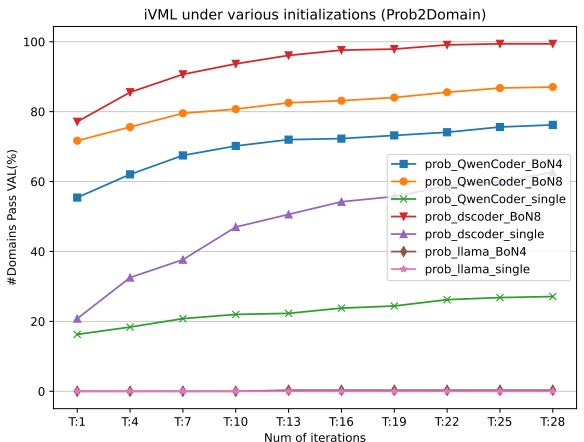

Figure 4: The performance of iVML on NL2Domain tasks across different initialization settings.

Figure 5: The performance of iVML on Prob2Domain tasks across different initialization settings.

| Model | Setting | Correct Rate (%) |
|---|---|---|
| Gemma 1.1 IT 2B | Zero-shot | 0.00 |
| | Fine-tuned | 94.21 |
| Gemma 1.1 IT 7B | Zero-shot | 0.00 |
| | Fine-tuned | 98.79 |
| Mistral v0.3 Instruct 7B | Zero-shot | 0.01 |
| | Fine-tuned | 99.00 |
| GPT-4o | Zero-shot | 35.12 |
| Ours (Qwen2.5-Coder-7B) | BoN-16 | 99.24 |
| Ours (Qwen2.5-Coder-7B) | iVML-1-BoN-16 | 99.60 |

Table 4: Performance comparison of different models on PDDL problem generation

to structured logical reasoning during pretraining, a deficiency in its pretraining knowledge that test-time compute scaling methods (*e.g.*, iVML and BoN) cannot effectively address.

## 4.5 PDDL Problem Generation

**Experiment settings.** This section investigates the effectiveness of our approach while generalizing it to PDDL problem synthesis. In contrast to the PDDL domain, which outlines the general framework or environment defined for planning tasks, the PDDL problem defines a specific instance of the planning task within that domain. This involves defining two main components (1) Initial state: the starting state of the world, defined by the predicates that are true initially, and (2) Goal state: The objective that the planner aims to achieve. We adopt the Planetarium [61] benchmark, which evaluates LLMs' capacity to generate precise PDDL problems from natural language descriptions. These tasks are challenging due to the lack of planning background knowledge and the complex context described by the problem. The evaluation methods outlined in [61] test LLMs in both zero-shot and fine-tuned settings. The baselines being evaluated include GPT-4, Gemma 1.1 IT models [41] with 2B and 7B parameters, as well as Mistral v0.3 Instruct (7B) [23]. Importantly, during PDDL problem generation, no pre-existing PDDL domain is provided as input—only natural language descriptions guide the synthesis process.

**Main results**. The results are presented in Table 4. We adopt the correct rate defined in Planetarium as the evaluation metric. This metric is defined as the percentage of generated problems whose goals correctly match the ground-truth goals. Planetarium fine-tuned Gemma and Mistral on a training dataset containing 132,027 examples from the two-class PDDL problem code dataset, potentially raising overfitting concerns as Gemma's accuracy increased dramatically from near 0.0% to over 98.8%. Our method enhances the Qwen2.5-

| Model | Setting | Floortile | Barman | Tyreworld | Grippers | Termes | Blockworld |
|---|---|---|---|---|---|---|---|
| *LLM-as-Planner Methods* | | | | | | | |
| GPT-4o | Pass@1 | 0.0 | 6.7 | 0.0 | 23.8 | 4.8 | 4.8 |
| | self-critique | 10.0 | 13.3 | 0.0 | 33.3 | 0.0 | 14.2 |
| | Pass@8 | 13.3 | 33.3 | 45.0 | 45.0 | 10.0 | 23.8 |
| o1-mini | Pass@1 | 5.3 | 33.3 | 50.0 | 57.1 | 23.8 | 38.1 |
| | self-critique | 5.3 | 33.3 | 35.0 | 61.9 | 23.8 | 47.6 |
| | Pass@8 | 0.0 | 33.3 | 70.0 | 61.9 | 52.4 | 23.1 |
| o1-preview | Pass@1 | 0.0 | 13.3 | 33.3 | 38.1 | 0.0 | 4.7 |
| | self-critique | 5.0 | 6.7 | 35.0 | 33.3 | 4.7 | 9.5 |
| | Pass@8 | 33.3 | 33.3 | 85.0 | 66.7 | 19.0 | 33.3 |
| *Oracle Interaction Methods* | | | | | | | |
| Qwen2.5-7B-Coder | Oracle Interaction | 0.0 | - | - | 0.0 | 100.0 | 100.0 |
| *Our Methods (PDDL as the Planning Representation)* | | | | | | | |
| Qwen2.5-7B-Coder | BoN-4 | 0.0 | 0.0 | 0.0 | 100.0 | 81.0 | 9.5 |
| Qwen2.5-7B-Coder | BoN-16 | 100.0 | 100.0 | 100.0 | 100.0 | 100.0 | 0.0 |
| Qwen2.5-7B-Coder | BoN-4-iVML-5 | 0.0 | 100.0 | 100.0 | 100.0 | 100.0 | 71.4 |
| Qwen2.5-7B-Coder | BoN-16-iVML-5 | 100.0 | 100.0 | 100.0 | 100.0 | 100.0 | 81.0 |

Table 5: PDDL as the planning representation methods vs. LLM-as-Planner methods. The evaluation metric is plan accuracy. Our PDDL-based method uses the Fast Downward system [19] for heuristic search and plan validation.

Coder (7B) model through test-time scaling techniques, achieving a 99.24% correctness rate with BoN-16 sampling. This improves further to 99.60% when combining iVML-1 with BoN-16 for solution initialization.

**Comparison between SFT and iVML**. The results in Table 4 provide a comparison between supervised finetuning (SFT) and iVML. This reveals three key advantages of our method (listed as follows). (1) *Better preservation of pretrained knowledge*: Unlike SFT's static alignment to fixed data distributions, iVML enforces structured reasoning priors through in-context learning, enabling adaptation to diverse problem constraints without triggering catastrophic forgetting of pretrained knowledge. (2) *Refinement through in-context optimization*: Building upon BoN's high-quality initialization, iVML performs in-context instance optimization to correct subtle errors, elevating correctness from 99.24% (BoN-16) to 99.60% (iVML-1-BoN-16). (3) *Computational efficiency*: iVML requires significantly less compute than SFT while demonstrating strong performance on complex reasoning tasks, including those in long-tail domains.

### 4.6 Comparison to LLM-as-Planner Methods

In the previous sections, we demonstrated that iVML enhances LLMs' ability to generate high-quality PDDL-based world models. In this section, we compare our method, which utilizes synthesized PDDL domains as world models, with LLMs-as-a-Planner methods that use natural language for world modeling and planning. A detailed description of the tested planning cases is provided in Appendix C.

**Experimental settings**. LLMs often hallucinate for tasks such as generating feasible or optimal plans, understanding the planning problem, and strictly following the rules, particularly in complex planning problems (*e.g.*, Termes and Barman) [44]. To investigate whether these hallucinations arise from limitations in prompt engineering, we provide LLMs with explicit instructions on the rules they must follow and require them to verify rule compliance at each step of the planning process. Additionally, we employ two strategies to reduce uncertainty in LLM-generated plans: (1) Introducing the Pass@8 metric to evaluate the probability that at least one of the top 8-generated plans is correct, and (2) Allowing LLMs to self-evaluate their plans and refine them based on these assessments. The baseline implementations of LLM-as-Planner methods are based on GPT-4o, o1-mini, and o1-preview, respectively.

**Discussion on LLM-as-Planner methods**. The observations are as follows: (1) *rule violation*: Despite explicitly informing LLMs to examine rule violation at each step, the generated plans still contain elements that inherently violate the predefined rules. For example, in the step 3 and 4 of Figure 2, o1 incorrectly places the block from pos-0-1 to pos-1-2, mistakenly assuming that they are neighboring positions. This action violates the rule governing the placement of blocks. (2) *incorrect state transition estimation*: The LLMs fail to accurately estimate state transitions. For instance, after moving the block from pos-1-1 to pos-1-2, o1 incorrectly assumes that the heights of both positions are 0, reflecting an inability to track state changes correctly. (3) *incorrect goal achievement estimation*: o1 attempts to achieve the goal in a manner that disregards all constraints and relies on flawed state estimations, which results in a plan that is not only incorrect but also violates the fundamental rules of the task. (4) *incorrect self-evaluation*: o1 fails to identify errors in its planning process, as it consistently assumes that its own responses are correct. This prevents it from correcting mistakes, further compounding the inaccuracies in its generated plans.

**PDDL-based method vs. LLM-as-Planner**. We present the numerical results in Table 5. For this experiment, we use planning accuracy as the metric. Specifically, for a given planning problem (*e.g.*, Termes), it measures the percentage of generated plans from the synthesized domains that exactly match the ground-truth reference plans. For classical planning problems, using natural language as a planning representation proves suboptimal. Even when equipped with self-critique capabilities, the o1-preview system achieves only 5.0 on Floortile, 6.7 on Barman, and 4.7 on Termes benchmarks. This limitation stems from natural language's inherent ambiguity and lack of formal precision required for precise planning representation. LLM-as-Planner methods do not perform planning; instead, they treat planning tasks as n-gram text completion problems, leading to plans that may lack feasibility, violate the constraints, or fail to accurately reflect the underlying problem structure. In contrast, our approach leverages PDDL representations to explicitly model state transitions. Through BoN-16-iVML-5 generation of high-quality world models combined with heuristic search algorithms (*i.e.*, A∗), our method solves nearly all tested instances across Floortile, Barman, and Termes domains. We compare our method with a PDDL-based approach [26] that uses the oracle environment to refine the answers. By contrast, our approach generates PDDL domains in an oracle-agnostic manner, leveraging test-time scaling to enhance scalability in scenarios without rule-based ground-truth signals. We found that our method (*e.g.*, BoN-4-iVML-5) achieves comparable performance to the oracle interaction method in the cases they considered, with both achieving 100.0 on Termes. However, our method outperforms the oracle interaction method on Floortile and Grippers.

## 5    Concluding Remarks and Current Limitations

Our work introduces a test-time scaling framework for automated PDDL synthesis that integrates best-of-N sampling with instance verbalized machine learning. This approach demonstrates that effectively scaling test-time computation of open-source LLMs can outperform state-of-the-art closed-source LLM planners, including OpenAI's o1-mini. Our hybrid method employs a two-phase optimization paradigm: (1) BoN initialization which generates diverse candidate solutions to explore critical regions of the search space, addressing the cold-start problem, where iVML begin with poor initialization in formal language synthesis. (2) iVML refinement which iteratively improves the BoN initial solutions through self-critique and natural language feedback, resolving logical inconsistencies and syntactic errors. Leveraging BoN's stochastic search to initialize iVML's refinement process, our method achieves faster convergence and higher-quality PDDL domains. The effectiveness of iVML in PDDL problem synthesis even surpasses models specifically fine-tuned for this task. The results show that our proposed test-time compute scaling approach can enhance LLMs' formal reasoning and planning capabilities. By generating PDDL-based symbolic world models, we enable explicit model-based planning with classical search algorithms (*e.g.*, A*), avoiding the error-prone state transitions inherent in direct LLM-as-planner approaches. Beyond PDDL synthesis, our work provides a general framework for scaling up test-time compute of LLMs for formal language synthesis.

The limitations of our work include: (1) *challenges in semantic verification for autoformalization*: Consistent with prior work in PDDL synthesis (*e.g.*, [17, 61, 42]), our evaluation relies on VAL [20] for syntax validation and plan verification. While VAL ensures syntactic correctness (*e.g.*, predicate arity, type consistency) and plan executability (*e.g.*, action preconditions and effects), it cannot detect semantic inconsistencies that violate domain intent or commonsense logic. This limitation parallels broader challenges in autoformalization,

where even formal mathematical proof [58] struggles to verify semantic alignment between informal specifications and formal outputs through compiler checking. (2) *simulation assumptions*: Our evaluation relies on an idealized simulation environment with two key assumptions. First, actions execute perfectly, with no execution misalignment. Second, the state space is fully observable, with no sensor noise or occlusions. These idealized conditions differ significantly from real-world robotic manipulation scenarios.

# 6 Broader Impact

The methodology presented in our paper carries significant implications for advancing the safe usage of human-centered AI agents. Unverifiable fully autonomous AI agents pose substantial risks to the security and safety of human society, as their capabilities are too strong while very difficult to control [5, 29]. Recent discourse in AI ethics and safety emphasizes that fully autonomous AI agents should not be developed without rigorous safeguards [29]. Current AI systems, particularly those relying on LLMs, exhibit significant hallucinations, such as hallucinations, rule violations, incorrect state transition estimations, and incorrect goal achievement assessments (see Figure 2). These issues stem from LLMs' reliance on n-gram text completion to model the world, a method that lacks logical coherence and a systemic understanding of real-world planning. On the other hand, self-verification mechanisms often fail to detect inherent errors in generated plans. In contrast, the scientist AI paradigm [5] advocates for a two-stage approach: first, learning a structured world model, and second, deriving rationally grounded inferences from it. Our method is similar to scientist AI [5] by employing test-time scaling to construct a symbolic world model before applying heuristic search to generate verifiable, rule-followed plans. By prioritizing symbolic reasoning over unstructured natural language generation, our approach mitigates risks associated with hallucinations and enhances the transparency of agent planning. We argue that this framework represents a critical step toward controllable, interpretable, and safe human-centered AI systems.

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

# Appendix

## Table of Contents

# A    Planning Problem Formulation

The classical planning problem [16] in artificial intelligence involves finding a sequence of actions that transition an agent from an initial state to a desired goal state within a deterministic and fully observable environment. It is formalized as a tuple $\langle \mathcal{S}, \mathcal{A}, T, s_0, G \rangle$, where: $\mathcal{S}$ is the set of all possible states; $\mathcal{A}$ is the set of all possible actions; $T : \mathcal{S} \times \mathcal{A} \rightarrow \mathcal{S}$ is the state transition function, specifying the outcome state resulting from applying an action in a state; $s_0 \in \mathcal{S}$ is the initial state; $G$ is the goal condition, a predicate over states. The objective is to find a sequence of actions $a_1, a_2, \ldots, a_n \in \mathcal{A}$ such that applying these actions successively transitions the system from $s_0$ to a state $s_n$ satisfying the goal condition $G$:

$$s_n = T(s_{n-1}, a_n) = T(T(\ldots T(T(s_0, a_1), a_2), \ldots, a_{n-1}), a_n)$$

with

$$s_n \models G$$

Previous works that adopt LLMs as planners estimate state transitions implicitly within language latent spaces, lacking explicit representations of the state space required for classical planning. This implicit representation can make it challenging to ensure consistency, validity, and completeness in the planning process. In contrast, our work leverages LLMs to generate explicit representations of the state space by creating PDDL domains. We utilize the generative capabilities of LLMs to produce formal PDDL models from high-level descriptions of the planning tasks. This approach bridges the gap between natural language specifications and formal planning models.

By generating PDDL domains using LLMs, we obtain: 1. explicit definitions of the set of states $\mathcal{S}$ through predicates and objects, 2. formal specifications of actions $\mathcal{A}$, including their preconditions and effects, 3. a deterministic state transition function $T$ derived from the action definitions, 4. abilities to cooperate with clearly defined initial state $s_0$ and goal condition $G$ in PDDL syntax. Thus the new objective under this background is: Given high-level descriptions of planning tasks, our objective is to leverage LLMs to create PDDL domains that can represent state transitions explicitly. By generating these explicit representations, we enable classical planning algorithms to efficiently search for plans using the defined state transitions during the planning process.

# B    STRIPS Formulation

The states are expressed through a set of predicates that describe the properties of objects in the environment. Each predicate represents a relationship or characteristic, such as `on(A, B)`, which indicates that object A is on top of object B. A state can be represented as:

$$S_0 = \{\text{on}(A, B), \text{clear}(C)\}$$

This representation includes relationships that capture the positions and statuses of the objects within the environment. Actions in PDDL are tightly bound to the representation of states through the use of preconditions and effects. Each action is defined by specifying what must be true in the current state for the action to be applicable (preconditions), as well as what changes in the state when the action is performed (effects). For example, consider an action `move(A, B, C)`, indicating that move the object A from B to C. The preconditions and effects could be defined as follows: Preconditions: $\text{Pre}(\text{move}(A, B, C)) = \{\text{on}(A, B), \text{clear}(C)\}$ and Effects: $\text{Eff}(\text{move}(A, B, C)) = \{\text{on}(A, C), \text{clear}(B)\}$. They indicate that for the action `move(A, B, C)` to be executed, object A must be positioned on B, and C must be clear of any objects. And after moving A from B to C, A is now on C, and B is clear. The transition can be formulated as: $T(S, A) \rightarrow S'$. State transitions occur according to a defined sequence of actions that an agent may take, leading to new configurations of the world. For instance if the action $\text{move}(A, B, C)$ on $S_0$, the transition can be represented as:

$$S_1 = T(S_0, \text{move}(A, B, C)) \rightarrow \{\text{on}(A, C), \text{clear}(B)\}$$

By repeatedly exploring all possible actions, we can form a state transition graph consisting of nodes (states) and directed edges (actions) that connect these nodes based on the actions that lead to different states. We continue this process by expanding the graph from the newly added nodes until the goal state is reached and added to the graph, or there are no more actions left to explore.

# C    Tasks in Case Study

**Barman.** The main goal of the Barman domain is to simulate the task of a bartender who prepares and serves cocktails by manipulating ingredients, tools, and glassware within a bar setting. In this scenario, the agent is tasked with creating a specific cocktail by following a series of actions that involve: 1. Identifying and dispensing the necessary ingredients required for the cocktail from the available dispensers. 2. Using bar tools such as shakers and shot glasses effectively, ensuring they are clean and suitable for use. 3. Coordinating the use of both hands to pick up and handle objects while ensuring that hands are free when needed and that objects are properly placed on surfaces like the bar counter when not in use. 4. Adhering to the proper sequence of steps for cocktail preparation, which includes dispensing ingredients into the shaker, mixing them, and then pouring the mixture into a shot glass. 5. Maintaining the correct state of all objects involved, such as keeping the shaker and glasses clean and empty before use, and updating their states appropriately as actions are performed. 6. Successfully preparing the cocktail and having it contained in the shot glass, thereby fulfilling the goal of serving the drink as intended.

**Gripper.** Gripper problem is designed to test the agent's ability to manage resources and plan actions in a scenario involving manipulating multiple objects across different locations. The agent, represented by one or more robots, must strategically perform the following tasks: Pick Up Balls: The agent must use its available grippers to pick up the balls from their initial locations. This requires careful hand management to ensure grippers are free and available when needed. Transport Balls: The robot needs to navigate between rooms to move balls to their specified target locations efficiently. This involves planning the correct sequence of moves and ensuring that the robot is in the correct room with the appropriate objects. Drop Balls: The agent must release the balls in the designated rooms. This requires ensuring that the robot's grippers are properly aligned and that the release actions are performed at the correct time. Manage Resources: Throughout the task, the agent must effectively manage both its grip and position within the environment, making sure that it follows constraints such as carrying capacity and room access.

**Tyreworld.** The main goal of the Tyreworld problem is to simulate the challenges of vehicle maintenance and tire management in a scenario where a vehicle may suffer random tire failures. The primary objectives include: 1. Replacing Flat Tires: The agent must effectively replace flat tires with intact ones on the vehicle's hubs. 2. Inflating Tires: The intact tires must be inflated before being mounted onto the vehicle. 3. Ensuring Secure Fastening: After replacing and inflating the tires, the nuts on the hubs must be securely tightened to ensure the wheels are safely attached. 3. Resource Management: The agent must manage limited resources (e.g., spare tires, tools like jacks and wrenches) strategically to minimize the risk of failure or being stranded. 4. Navigating Uncertainty: The agent must effectively plan actions while accounting for the possibility of tire failures and other uncertainties in the environment. 5. Reaching the Destination: Ultimately, the goal is to ensure the vehicle is properly equipped with intact, inflated tires, allowing it to continue its journey successfully. 6. The Tyreworld problem serves to test an agent's planning, resource management, and adaptability in unpredictable scenarios related to vehicle maintenance.

**Floor-tile.** The main goal of Floor-tile is to enable the robot to navigate the environment, manage its colors, and paint the tiles according to specific requirements. The robot must efficiently utilize its movements and actions to achieve its painting objectives while adhering to the constraints of tile occupancy and color availability. In Floor-tile, three object types are defined: robot, tile, and color. Initially, the robot is located on a specific tile while holding a color. It can move up, down, right, or left with different costs: moving up costs 3, while moving down, right, or left costs 1. The robot can paint a tile above or below for a cost of 2, and changing its color incurs a cost of 5.

**Termes.** In Termes, a robot operates in an environment to manage and manipulate blocks at different positions. The robot can perform several actions, including moving between adjacent positions, placing blocks onto stacks, removing blocks from stacks, and creating or destroying blocks at designated depot locations. Each position on the grid has a specific height, and the robot must respect these height constraints when moving or manipulating blocks. The robot can only carry one block at a time, and it must be at the same height level to move horizontally or at a height difference of one to move vertically. The goal is to efficiently use these capabilities to achieve specific block arrangements or configurations within the environment, adhering to the constraints of adjacency, height, and block availability.

## D Prompt Template for Chain-of-Thought

**Prompts for Our Methods**

You will be given a natural language description of a planning problem. Your task is to translate this description into PDDL domain code. This includes defining predicates and actions based on the information provided.

Information about the AI agent will be provided in the natural language description. Note that individual conditions in preconditions and effects should be listed separately. For example, "object1 is washed and heated" should be considered as two separate conditions "object1 is washed" and "object1 is heated". Also, in PDDL, two predicates cannot have the same name even if they have different parameters. Each predicate in PDDL must have a unique name, and its parameters must be explicitly defined in the predicate definition. It is recommended to define predicate names in an intuitive and readable way. Remember: Ignore the information that you think is not helpful for the planning task. You are only responsible for domain generation. Before you generate the concrete domain code, you should first generate a natural language thought about the meaning of each variable, and the step-by-step explaination of the domain code. Even if I didn't provide the exact name of the predicates and actions, you should generate them based on the information provided in the natural language description.

Template is:

### Thought:

predicates1: the name of predicate1, explanation of predictate1

...

predicaten: the name of predicaten, explanation of predictaten

action1: the name of action1, explanation of action

...

actionn: the name of action, explanation of action

<thought>

### Domain: "'pddl

The concrete pddl code for domain.pddl

Now its your time to generate the solution, you have to follow the format I provided above.

NL_Description: Natural language description of the planning domain

# E   Generated Domains

## Barman

**Domain.**

```
(define (domain barman)
  (:requirements :strips :typing)
  (:types hand level beverage dispenser container − object ingredient cocktail − beverage
shot shaker − container)
  (:predicates  (ontable ?c − container)
                (holding ?h − hand ?c − container)
                (handempty ?h − hand)
                (empty ?c − container)
                (contains ?c − container ?b − beverage)
                (clean ?c − container)
                (used ?c − container ?b − beverage)
                (dispenses ?d − dispenser ?i − ingredient)
                (shaker−empty−level ?s − shaker ?l − level)
                ......
  (:action grasp
                :parameters (?h − hand ?c − container)
                :precondition (and (ontable ?c) (handempty ?h))
                :effect (and (not (ontable ?c)) (not (handempty ?h)) (holding ?h ?c)))
  (:action leave ......
  (:action fill−shot ......
  (:action refill−shot ......
  (:action empty−shot ......
  (:action clean−shot ......
  (:action pour−shot−to−clean−shaker ......
  (:action pour−shot−to−used−shaker ......
  (:action empty−shaker ......
  (:action clean−shaker ......
  (:action shake ......
  (:action pour−shaker−to−shot ...... )
```

**Problem.**

```
(define (problem prob)
  (:domain barman)
  (:objects
        shaker1 − shaker left right − hand shot1 shot2 shot3 shot4 − shot ingredient1
        ingredient2 ingredient3 − ingredient cocktail1 cocktail2 cocktail3 − cocktail
        dispenser1 dispenser2 dispenser3 − dispenser l0 l1 l2 − level
)
  (:init
  (ontable shaker1)
  (ontable shot1) ......
  (clean shaker1) ......
  (empty shaker1) ......
  (cocktail−part1 cocktail1 ingredient1) ......
)
  (:goal
  (and (contains shot1 cocktail1) (contains shot2 cocktail3) (contains shot3 cocktail2)
```

**Plan.** (grasp left shot4) (fill-shot shot4 ingredient2 left right dispenser2) (pour-shot-to-clean-shaker shot4 ingredient2 shaker1 left l0 l1) (clean-shot shot4 ingredient2 left right) (fill-shot shot4 ingredient1 left right dispenser1) (pour-shot-to-used-shaker shot4 ingredient1 shaker1 left l1 l2) (refill-shot shot4 ingredient1 left right dispenser1) (leave left shot4) (grasp right shaker1) (shake cocktail3 ingredient1 ingredient2 shaker1 right left) (pour-shaker-to-shot cocktail3 shot2 right shaker1 l2 l1) (empty-shaker right shaker1 cocktail3 l1 l0) (clean-shaker right left shaker1) (leave right shaker1) (grasp left shot4) (pour-shot-to-clean-shaker shot4 ingredient1 shaker1 left l0 l1) (clean-shot shot4 ingredient1 left right) (fill-shot shot4 ingredient3 left right dispenser3) (pour-shot-to-used-shaker shot4 ingredient3 shaker1 left l1 l2) (refill-shot shot4 ingredient3 left right dispenser3) (leave left shot4) (grasp right shaker1) (shake cocktail1 ingredient1 ingredient3 shaker1 right left) (pour-shaker-to-shot cocktail1 shot1 right shaker1 l2 l1) (empty-shaker right shaker1 cocktail1 l1 l0) (clean-shaker right left shaker1) (leave right shaker1) (grasp right shot4) (pour-shot-to-clean-shaker shot4 ingredient3 shaker1 right l0 l1) (clean-shot shot4 ingredient3 right left) (fill-shot shot4 ingredient2 right left dispenser2) (grasp left shaker1) (pour-shot-to-used-shaker shot4 ingredient2 shaker1 right l1 l2) (leave right shot4) (shake cocktail2 ingredient2 ingredient3 shaker1 left right) (pour-shaker-to-shot cocktail2 shot3 left shaker1 l2 l1) ; cost = 36 (unit cost)

## BlockWorld

**Domain.**

```
(define (domain blocksworld)
  (:requirements :strips :equality)

  (:predicates
    (clear ?x)
    (on-table ?x)
    (arm-empty)
    (holding ?x)
    (on ?x ?y))

  (:action pickup
    :parameters (?ob)
    :precondition (and (clear ?ob) (on-table ?ob) (arm-empty))
    :effect (and (holding ?ob) (not (clear ?ob)) (not (on-table ?ob))
    (not (arm-empty))))

  (:action putdown
    :parameters (?ob)
    :precondition (and (holding ?ob))
    :effect (and (clear ?ob) (arm-empty) (on-table ?ob)
    (not (holding ?ob))))

  (:action stack
    :parameters (?ob ?underob)
    :precondition (and (clear ?underob) (holding ?ob))
    :effect (and (arm-empty) (clear ?ob) (on ?ob ?underob) (not (clear ?underob))
    (not (holding ?ob))))

  (:action unstack
    :parameters (?ob ?underob)
    :precondition (and (on ?ob ?underob) (clear ?ob) (arm-empty))
    :effect (and (holding ?ob) (clear ?ob) (not (on ?ob ?underob)) (not (clear
    ?ob)) (not (arm-empty))) ))
```

**Problem.**

```
(define (problem BW-rand-12)
    (:domain blocksworld)
    (:objects b1 b2 b3 b4 b5 b6 b7 b8 b9 b10 b11 b12 )
    (:init
        (arm-empty)(on-table b1)(on b2 b5)(on b3 b8)(on b4 b12)(on b5 b7)
        (on b6 b1)(on b7 b10)
        (on-table b8)(on-table b9)(on b10 b11)(on-table b11)(on b12 b9)
        (clear b2)(clear b3)(clear b4)(clear b6))
    (:goal
        (and (on b5 b10)(on b6 b12)(on b7 b4)(on b8 b3)(on b9 b2)(on b10 b8)
        (on b11 b7)(on b12 b11))))
```

**Plan.**

(unstack b3 b8) (putdown b3) (pickup b8) (stack b8 b3) (unstack b2 b5) (putdown b2) (unstack b4 b12) (putdown b4) (unstack b5 b7) (stack b5 b2) (unstack b7 b10) (stack b7 b4) (unstack b10 b11) (stack b10 b8) (pickup b11) (stack b11 b7) (unstack b12 b9) (stack b12 b11) (unstack b5 b2) (stack b5 b10) (unstack b6 b1) (stack b6 b12) (pickup b9) (stack b9 b2) ; cost = 24 (unit cost)

## Termes

**Domain**.

```
(define (domain termes)
    (:requirements :typing :negative-preconditions)
    (:types
        numb - object
        position - object)
    (:predicates
        (height ?p - position ?h - numb)
        (at ?p - position)
        (has-block)
        (SUCC ?n1 - numb ?n2 - numb)
        (NEIGHBOR ?p1 - position ?p2 - position)
        (IS-DEPOT ?p - position))
    (:action move
        :parameters (?from - position ?to - position ?h - numb)
        :precondition (and (at ?from)(NEIGHBOR ?from ?to)(height ?from ?h)(height ?to ?h))
        :effect (and (not (at ?from))(at ?to) ) )
    (:action move-up
        :parameters (?from - position ?hfrom - numb ?to - position ?hto - numb)
        :precondition (and(at ?from)(NEIGHBOR ?from ?to)(height ?from ?hfrom)(height
        ?to ?hto)(SUCC ?hto ?hfrom))
        :effect (and(not (at ?from))(at ?to)))

    (:action move-down ...
    (:action place-block ...
    (:action remove-block ...
    (:action create-block ...
    (:action destroy-block ...
)
```

**Problem**.

```
(define (problem termes-00038-0036-4x3x3-random_towers_4x3_3_1_3)
(:domain termes)
; termes-00038-0036-4x3x3-random_towers_4x3_3_1_3
; Initial state:
;  0   0   R0D  0
;  0   0   0    0
;  0   0   0    0
; Goal state:
;  0   0   0    0
;  0   0   0    0
;  0   3   0    0
; Maximal height: 3
(:objects
    n0 - numb......
    pos-0-0 - position......
)
(:init
    (height pos-0-0 n0)......
    (at pos-2-0)
    (SUCC n1 n0)......
    (NEIGHBOR pos-0-0 pos-1-0)......
    (IS-DEPOT pos-2-0)
)
(:goal
(and (height pos-0-0 n0)  ......  (not (has-block)))))
```

- - - - - - - - - - - - - - - - - - - - - - - - - - - - - - - - - - - - - - - - - - - - -

**Plan**.
(unstack b3 b8) (putdown b3) (pickup b8) (stack b8 b3) (unstack b2 b5) (putdown b2) (unstack b4 b12) (putdown b4) (unstack b5 b7) (stack b5 b2) (unstack b7 b10) (stack b7 b4) (unstack b10 b11) (stack b10 b8) (pickup b11) (stack b11 b7) (unstack b12 b9) (stack b12 b11) (unstack b5 b2) (stack b5 b10) (unstack b6 b1) (stack b6 b12) (pickup b9) (stack b9 b2) ; cost = 24 (unit cost)

## Floor-tile

**Domain.**

```
(define (domain floor-tile)
  (:requirements :typing :action-costs)
  (:types robot tile color - object)

  (:predicates
    (robot-at ?r - robot ?x - tile)
    (up ?x - tile ?y - tile)
    (down ?x - tile ?y - tile)
    (right ?x - tile ?y - tile)
    (left ?x - tile ?y - tile)

    (clear ?x - tile)
    (painted ?x - tile ?c - color)
    (robot-has ?r - robot ?c - color)
    (available-color ?c - color)
    (free-color ?r - robot))

  (:functions (total-cost))
  (:action change-color
    :parameters (?r - robot ?c - color ?c2 - color)
    :precondition (and (robot-has ?r ?c) (available-color ?c2))
    :effect (and (not (robot-has ?r ?c)) (robot-has ?r ?c2) (increase (total-cost) 5)))
  (:action paint-up
    :parameters (?r - robot ?y - tile ?x - tile ?c - color)
    :precondition (and (robot-has ?r ?c) (robot-at ?r ?x) (up ?y ?x) (clear ?y))
    :effect (and (not (clear ?y)) (painted ?y ?c) (increase (total-cost) 2))
  )
  (:action paint-down...
  (:action up ...
  (:action down ...
  (:action right ...
  (:action left ...
```

**Problem.**

```
(define (problem p03-432)
  (:domain floor-tile)
  (:objects tile_0-1 tile_0-2 tile_0-3 ......tile_4-1 tile_4-2 tile_4-3 - tile
            robot1 robot2 - robot
            white black - color
)
  (:init
    (= (total-cost) 0)
    (robot-at robot1 tile_2-3)
    (robot-has robot1 white)
    (robot-at robot2 tile_1-1)
    (robot-has robot2 black)
    (available-color white)
    (available-color black)
    (clear tile_0-1) ......
    (up tile_1-1 tile_0-1) ......
    (down tile_0-1 tile_1-1) ......
    (right tile_0-2 tile_0-1) ......
    (left tile_0-1 tile_0-2) ......)
  (:goal (and
    (painted tile_1-1 white)
    (painted tile_1-2 black) ......))
  (:metric minimize (total-cost)))
```

**Plan.** (up robot1 tile_2-3 tile_3-3) (left robot1 tile_3-3 tile_3-2) (paint-up robot1 tile_4-2 tile_3-2 white) (up robot2 tile_1-1 tile_2-1) (down robot1 tile_3-2 tile_2-2) (up robot2 tile_2-1 tile_3-1) ...... ; cost = 54 (general cost)

## Tyreworld

**Domain**.

```
(define (domain tyreworld)
  (:types obj − object
          tool wheel nut − obj
          container hub − object)

  (:predicates (open ?x) (closed ?x) (have ?x) (in ?x ?y) (loose ?x ?y)
        (tight ?x ?y)
        (unlocked ?x) (on−ground ?x) ......
  (:action open
    :parameters (?x − container)
    :precondition (and (unlocked ?x) (closed ?x))
    :effect (and (open ?x) (not (closed ?x))))
  (:action close
    :parameters (?x − container)
    :precondition (open ?x)
    :effect (and (closed ?x) (not (open ?x))))
  (:action fetch
    :parameters (?x − obj ?y − container)
    :precondition (and (in ?x ?y) (open ?y))
    :effect (and (have ?x) (not (in ?x ?y))))
  (:action put−away ......
  (:action loosen ......
  (:action tighten ......
  (:action jack−up ......
  (:action jack−down ......
  (:action undo ......
  (:action do−up ......
  (:action remove−wheel ......
  (:action put−on−wheel ......
  (:action inflate ......
```

**Problem**.

```
(define (problem tyreworld−1)
(:domain tyreworld)
(:objects  wrench jack pump − tool the−hub1 − hub nuts1 − nut
    boot − container r1 w1 − wheel)
(:init (in jack boot) (in pump boot) (in wrench boot) (unlocked boot)
    (closed boot) (intact r1) (in r1 boot) (not−inflated r1) (on w1 the−
    hub1) (on−ground the−hub1) (tight nuts1 the−hub1) (fastened the−hub1))
(:goal
    (and (on r1 the−hub1) (inflated r1) (tight nuts1 the−hub1) (in w1 boot)
    (in wrench boot) (in jack boot) (in pump boot) (closed boot))))
```

- - - - - - - - - - - - - - - - - - - - - - - - - - - - - - - - - - - - - - - -

**Plan**.

(open boot) (fetch r1 boot) (fetch wrench boot) (fetch jack boot) (loosen nuts1 the-hub1) (jack-up the-hub1) (undo nuts1 the-hub1) (remove-wheel w1 the-hub1) (put-away w1 boot) (put-on-wheel r1 the-hub1) (do-up nuts1 the-hub1) (jack-down the-hub1) (put-away jack boot) (tighten nuts1 the-hub1) (put-away wrench boot) (fetch pump boot) (inflate r1) (put-away pump boot) (close boot)
; cost = 19 (unit cost)

## F  Prompts for LLM-as-Planner Methods

---

**Prompts for o1 on Termes**

**Problem Description**. You control a robot that can take the following actions to build complex structures.
Move from a position to another. The new position and the old position must be at the same height.
Move up from a position to another, and the height at the new position is one block higher than the old position.
Move down from a position to another, and the height at the new position is one block lower than the old position.
Place a block at a neighboring position from the robot's current position. The robot must have a block. The current height at the robot's position and the block's position must be the same. A block cannot be placed at the depot. The height at the block's position will be one block higher than the current height.
Remove a block at a neighboring position from the robot's current position. The robot must not have a block. A block cannot be removed from the depot. The current height at the robot's position must be the same as the new height at the block's position. The new height at the block's position will be one block lower than the current height.
Create a block at the depot. The robot will have the block.
Destroy a block at the depot. The robot must have a block. Now consider a planning problem. The problem description is: The robot is on a grid with 4 rows and 3 columns. pos-0-0 pos-0-1 pos-0-2 pos-1-0 pos-1-1 pos-1-2 pos-2-0 pos-2-1 pos-2-2 pos-3-0 pos-3-1 pos-3-2 The robot is at pos-2-0. The depot for new blocks is at pos-2-0. The maximum height of blocks is 3. Your goal is to build blocks so that the height at pos-1-2 is 3. Rule: You cannot have an unplaced block at the end. Examine whether you follow the rule at each step! Can you provide an optimal plan, in the way of a sequence of behaviors, to solve the problem? And what is the final optimal cost?

---

# G State-based graph for Termes

Figure 6 shows the planning graph that contains explicit state transition during planning for Termes.

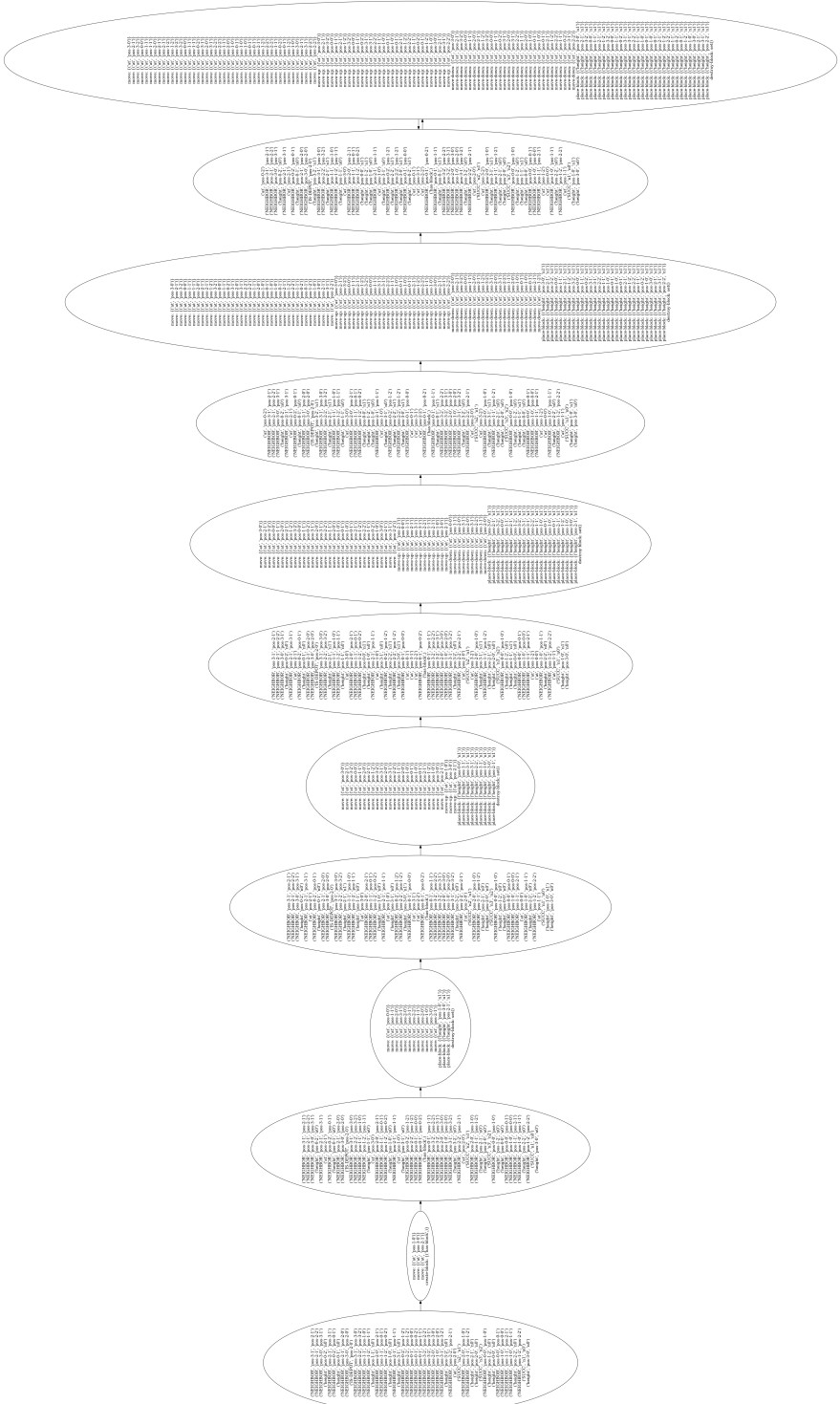

Figure 6: The planning graph for Termes

## H  The Prompt of Our Methods on Termes

> **Prompts for Our Methods on Termes**
>
> You will be given a natural language description of a planning problem. Your task is to translate this description into PDDL domain code. This includes defining predicates and actions based on the information provided.
>
> Information about the AI agent will be provided in the natural language description. Note that individual conditions in preconditions and effects should be listed separately. For example, "object1 is washed and heated" should be considered as two separate conditions "object1 is washed" and "object1 is heated". Also, in PDDL, two predicates cannot have the same name even if they have different parameters. Each predicate in PDDL must have a unique name, and its parameters must be explicitly defined in the predicate definition. It is recommended to define predicate names in an intuitive and readable way. Remember: Ignore the information that you think is not helpful for the planning task. You are only responsible for domain generation. Before you generate the concrete domain code, you should first generate a natural language thought about the meaning of each variable, and the step-by-step explaination of the domain code. Even if I didn't provide the exact name of the predicates and actions, you should generate them based on the information provided in the natural language description.
> Template is:
> ### Thought: predicates1: the name of predicate1, explanation of predictate1 ... predicaten: the name of predicaten, explanation of predictaten action1: the name of action1, explanation of action1 ... actionn: the name of action, explanation of actionn <thought>
> ### Domain: "'pddl The concrete pddl code for domain.pddl
> Now its your time to generate the solution, you have to follow the format I provided above.
> NL_Description:
> You control a robot that can take the following actions to build complex structures.
> Move from a position to another. The new position and the old position must be at the same height. – pddl action name: move
> Move up from a position to another, and the height at the new position is one block higher than the old position. – pddl action name: move-up
> Move down from a position to another, and the height at the new position is one block lower than the old position. – pddl action name: move-down
> Place a block at a neighboring position from the robot's current position. The robot must have a block. The current height at the robot's position and the block's position must be the same. A block cannot be placed at the depot. The height at the block's position will be one block higher than the current height. – pddl action name: place-block
> Remove a block at a neighboring position from the robot's current position. The robot must not have a block. A block cannot be removed from the depot. The current height at the robot's position must be the same as the new height at the block's position. The new height at the block's position will be one block lower than the current height. – pddl action name: remove-block
> Create a block at the depot. The robot will have the block. – pddl action name: create-block
> Destroy a block at the depot. The robot must have a block. – pddl action name: destroy-block
> An example problem PDDL file to the domain is:
> "'pddl (define (problem prob) (:domain termes) ; Initial state: ; 0 0 R0D ; 0 0 0 ; 0 0 0 ; Goal state: ; 0 0 0 ; 0 1 0 ; 0 0 0 ; Maximal height: 1 (:objects n0 - numb n1 - numb pos-0-0 - position pos-0-1 - position pos-0-2 - position pos-1-0 - position pos-1-1 - position pos-1-2 - position pos-2-0 - position pos-2-1 - position pos-2-2 - position ) (:init (height pos-0-0 n0) (height pos-0-1 n0) (height pos-0-2 n0) (height pos-1-0 n0) (height pos-1-1 n0) (height pos-1-2 n0) (height pos-2-0 n0) (height pos-2-1 n0) (height pos-2-2 n0) (at pos-2-0) (SUCC n1 n0) (NEIGHBOR pos-0-0 pos-1-0) (NEIGHBOR pos-0-0 pos-0-1) (NEIGHBOR pos-0-1 pos-1-1) (NEIGHBOR pos-0-1 pos-0-0) (NEIGHBOR pos-0-1 pos-0-2) (NEIGHBOR pos-0-2 pos-0-1) (NEIGHBOR pos-1-0 pos-0-0) (NEIGHBOR pos-1-0 pos-2-0) (NEIGHBOR pos-1-0 pos-1-1) (NEIGHBOR pos-1-1 pos-0-1) (NEIGHBOR pos-1-1 pos-2-1) (NEIGHBOR pos-1-1 pos-1-0) (NEIGHBOR pos-1-1 pos-1-2) (NEIGHBOR pos-1-2 pos-0-2) (NEIGHBOR pos-1-2 pos-2-2) (NEIGHBOR pos-1-2 pos-1-1) (NEIGHBOR pos-2-0 pos-1-0) (NEIGHBOR pos-2-0 pos-2-1) (NEIGHBOR pos-2-1 pos-1-1) (NEIGHBOR pos-2-1 pos-2-0) (NEIGHBOR pos-2-1 pos-2-2) (NEIGHBOR pos-2-2 pos-1-2) (NEIGHBOR pos-2-2 pos-2-1) (IS-DEPOT pos-2-0) ) (:goal (and (height pos-0-0 n0) (height pos-0-1 n0) (height pos-0-2 n0) (height pos-1-0 n0) (height pos-1-1 n1) (height pos-1-2 n0) (height pos-2-0 n0) (height pos-2-1 n0) (height pos-2-2 n0) (not (has-block)) ) ) )

# I   Human in the Loop Experiment

We conducted human-in-the-loop experiments to refine the process of writing PDDL (Planning Domain Definition Language) domain code with the interaction between artificial intelligence and humans. The experiment involved graduate students majoring in AI and robotics, focusing on evaluating the effectiveness of human-AI collaboration in generating accurate and semantically meaningful PDDL domain files that strictly adhere to given specifications.

The planning domain selected for this study was Termes, which necessitated the translation of robot actions into PDDL. These actions included horizontal and vertical movements, block placement and removal, and depot management within a simulated environment.

In the initial phase, participants interacted directly with an AI agent by providing prompts based on descriptions of robot actions. While the AI-generated PDDL code passed basic validation, it often lacked a proper definition of action preconditions or the accurate use of predicates, such as the "SUCC" predicate, which denotes an ordered relationship between items. In response to these limitations, the students refined their prompting strategy by incorporating more structured instructions and examples, resulting in improved outcomes.

Simultaneously, students manually coded PDDL domain files, utilizing the AI agent to ensure grammatical correctness. This approach facilitated the creation of logically comprehensive domain files, though several iterations were still required to achieve successful validation.

Through this iterative process, students provided critical insights into the current strengths and weaknesses of AI in comprehending complex logical structures and semantic nuances within specialized domains like PDDL. Their findings underscored the challenges associated with writing precise PDDL code and emphasized the need for an automated pipeline to facilitate PDDL domain synthesis.

