# OpenReview forum: "Generating Symbolic World Models via Test-time Scaling of Large Language Models"
_TMLR — Accepted by TMLR_

### Review · Reviewer_KXRw · 2025-02-16

**Summary Of Contributions:**

The paper proposes a novel prompting, training-free framework for LLMs to address complex planning problems with rules written in natural language or code. To do this, the proposed solution transforms them into domains of Planning Domain Definition Language (PDDL), which serves as a world model and can then be solved by classic searching algorithms. The framework consists of two steps: 1) best-of-N initialization where the initialized domain codes with the highest log probabilities are chosen as the starting points, and 2) iterative critique and improvement via multi-agent conversations. The proposed method is evaluated on many planning tasks (much more than existing works in this field) and is proved to be effective, generalizable, and also applicable for PDDL problem generation.

**Audience:**

Yes

**Broader Impact Concerns:**

1. The paper does not contain a broader impact section. While I think the contribution of this paper does not have significant ethical concern, I still encourage the authors to add a broader impact section on considering how LLMs that automates planning process could affect human society.

2. In Appendix I, a human-involved experiment is mentioned where participants interacts with AI agents. The human participation details of this paper is unclear. For example, how many people are involved in this experiment, and are they researchers themselves or hired people? If it is the latter, are they properly paid and informed? As those details are not specified in the paper, the paper contains a potential ethical risk.

**Claims And Evidence:**

Yes

**Requested Changes:**

**Major Changes**

1. The paper seems to be unclear on how accuracy is defined for domain synthesis. More specifically, how is a domain considered to be "accurate" in Fig. 3? Also, in the caption of Tab. 5 in Sec. 4.5, the author explained that "the comparison uses plan accuracy as the evaluation critic". What is the definition of "plan accuracy" here? Is it the success rate of classic searching algorithms?

2. It is quite surprising that the performance continue to grow even after 80 epochs in Fig. 3 for 7B models, which means the generated domain code keeps getting improved over 80 modifications. Such result indicates that the model has potential to address the task very well (almost perfect as suggested by the total number of domains in Tab. 1), but for some reason its ability must be extracted via a very large number of reflections (essentially self-reflection as the backbone model for $f_{\text{opt}}$ and $f_{\text{learner}}$ are the same). Could the author add an ablation / visualization throughout all the steps to explain why? Is it because the generated code is very long such that many errors?

3. The planning graph (Fig. 6) in Appendix G is confusing and too small to examine. Could the authors explain more about the planning graph?

4. The first step of the framework "generates multiple diverse and high-quality initializations" (quote from end of Sec. 3), and iVML refinement "optimizes these initial solutions". The paper seems to indicate that the multiple initial solutions chosen (K solutions in Sec. 3.2) are optimized by iVML in parallel. This is also indicated in the caption of Tab. 2 and "experimental settings" in Sec. 4.6,  where the authors write "pass@8 metric ... at least one of the top 8-generated plans is correct". My questions are:

a) Could the authors clarify how the single numbers in of Tab. 2 and Fig. 3 are acquired?

b) The method seems to require $O(TN)$ LLM calls, which could be a potential shortcoming. Could the authors report the numbers of token consumption for their proposed method?

5. The proposed method are not sufficiently compared to non-trivial prompt-based baselines, e.g., reflexion [1] (which is tested in Sec. 4.5 only), LATS [2], RAP [3] with generated code as "action". I suggest the authors to add prompt-based baselines to strengthen the evaluation of their work.

**Minor Changes**

1. In the 4th line of Sec. 5, add a white space between sentences in "... o1-mini.Our hybrid ...".

2. before prompt template for $f_{\text{opt}}(\cdot)$: "best-of-N sampleing" -> "best-of-N sampling".

3. The author could add more discussion on PDDL-related work, and briefly mention the problem formulation in the main paper to make it more self-contained.

4. There are too much text in Fig. 2; it would be better if the author could summarize the contents, highlight the incorrect part by o1 and redraw Fig. 2.

**References**

[1] N. Shinn et al. Reflexion: Language Agents with Verbal Reinforcement Learning. In NeurIPS, 2023.

[2] A. Zhou et al. Language Agent Tree Search Unifies Reasoning Acting and Planning in Language Models. In ICML, 2024.

[3] S. Hao et al. Reasoning with Language Model is Planning with World Model. In EMNLP, 2023.

**Strengths And Weaknesses:**

**Strengths**

1. The high-level idea of this work is simple, effective, and clearly conveyed in the paper. The authors propose a simple two-step training-free framework, with the first step being best-of-N and second step being iterative reflection, which is clearly introduced by Fig. 1 and Sec. 3 of this paper. Tab. 1 clearly shows the advantage of this method compared to prior work in PDDL synthesis methods.

2. The empirical evaluation is sufficient and clearly shows that the proposed method can improve the performance of LLMs solving planning problems. It is promising to see that even a 7B model can achieve near-perfect accuracy on many different domains as shown in the paper.

3. The authors provide many details on prompts of this work in the appendix, which makes the work more reproducible and convincing.

**Weaknesses**

See "requested changes" for details. Briefly summarizing:

1. Some details of the paper is unclear and requires explanation / extra ablations (major changes 1, 2, 3, 4a; broader impact concerns);

2. The proposed method is potentially inefficient (major changes 4b);

3. The paper are not sufficiently compared to non-trivial prompt-based baselines (major changes 5).

---

> ### Author Response · Authors · 2025-03-18
> **Response from the authors (Part 1)**
>
> We want to sincerely thank the reviewer for the constructive and thorough comments and suggestions on our work. We take every comment seriously and hope our response can address the reviewer’s concerns. If there are any remaining questions, we are more than happy to address them.
>
> ---
>
> ## MC1: Unclear Accuracy Metric
>
> Thank you for raising these important questions. We clarify our evaluation metrics as follows:
>
> 1. Domain Synthesis Accuracy (Table 2 and Fig. 3)
>
>     A synthesized domain was considered “accurate” if it passed all syntax and semantic validation checks using the [VAL](https://github.com/KCL-Planning/VAL) toolchain, following the previous works [1][2][3]. VAL is able to check the syntax errors in the pddl domains, e.g. invalid predicates parameters types, invalid predicate name, invalid predicate format, incorrect predicate usage, etc [3].
>
>     Our accuracy metric represents the percentage of domains that fully pass VAL validation.
>
>     Note that even passing the VAL checking is not a simple task, (i.e. Llama3.1-Instruct-70B only achieves 1.1% and 0.0% VAL passing rate on NL2Domain and Prob2Domain tasks).
>
>     We have changed the “accurate" in Fig. 3 to the “#Domains pass VAL” and clarified the definition of the metric used in PDDL domains generation in the Sec. 4.1.
>
> 2. PDDL problem success rate (Tab. 4)
>
>     Following the original benchmark Planetarium[4], we use the percentage of correct generated problems (i.e., the generated symbolic goals match with the GT goals). We have clarified the metric used in PDDL problem generation in Sec 4.5.
>
> 3. Planning Accuracy (Tab. 5)
>
>     For a given planning problem (e.g., Termes), it measures the percentage of generated plans from the synthesized domains that exactly match the ground-truth reference plans. See Sec. 4.6.
>
>
> ## MC2 and MC4 (b): concern about large token consumption and O(TN) complexity
>
> ### MC2: Token consumption
>
> Thank you for raising the concern about the potential shortcoming of our methods that may consume a large amount of tokens and LLMs call.
>
> We clarify that:
>
> 1. PDDL domain code contains lots of symbols, and LLMs need to refine the code many times to keep the internal logical coherence.
> 2. It is common for test-time scaling methods to consume a large amount of LLM tokens and LLM calls, as we show in the following experiment, our method is actually more efficient than other test-time scaling methods (e.g., LATS[5] that uses MCTS).
> 3. Most of the generated domains (85.2%) can pass the validation tool checking before 10 iVML iterations, only less 14.8% needs to undergo further improvement of iVML.
>
> To prove the first and the second claim, we conduct an additional experiment that uses Claude-3.5-Sonnet  to evaluate the semantic alignment between generated domains and their corresponding natural language descriptions. The domains are under different difficulties (i.e., the domain that consumes more tokens is regarded to be more difficult). We compare iVML with LATS(MCTS) to illustrate that the large token consumption is a common issue in test-time scaling methods. iVML still outperforms LATS on all the domains we considered in this experiment.
>
> We use the following prompt to check the semantic alignment:
>
> ```markdown
> You are a PDDL assistant, you will be given a natural language description of a planning problem and a generated PDDL domain.
> Your task is to judge the semantic correctness of the PDDL domain, to judge how much the high-level semantics of the PDDL domain match the natural language description of the planning problem.
> Use 0-10 to score the pddl domain, 0 is the worst, 10 is the best.
> In the end, please include your score in 【】.
>
> Here is the natural language description of the planning problem:
> {problem_description}
>
> Here is the generated PDDL domain:
> {pddl_domain}
>
> Please give your score in 【】.
> ```
>
> | **Planning Domain**  | **iVML Claude Score** | **iVML Token Consumption**  | **LATS Claude Score** | **LATS Token Consumption** |
> | --- | --- | --- | --- | --- |
> | package-delivery | 9  | 72k | 6 | 7k |
> | rover-exploration | 8   | 1.2M | 3 | 2M |
> | transport | 8 | 456k | 5 | 1.5M |
> | grippers | 10 | 80k | 8 | 7k |

---

> > ### Author Response · Authors · 2025-03-18
> > **Response from the authors (Part 2)**
> >
> > ### MC4(b): LLM Calls Complexity
> >
> > Our method is more efficient than the LATS approach mentioned by the reviewer [6]. The computational complexity of LATS is O(P * T * N), where:
> >
> > - P represents the number of pass@k evaluations
> > - T denotes the maximum iteration count
> > - N corresponds to the Best-of-N sampling parameter in LATS' implementation
> >
> > In contrast, our proposed method achieves a base complexity of O(TN). Furthermore, we leverage GPU parallelism (i.e. utilizing 8 H100 GPUs and tensor parallelism) to reduce the effective complexity to O(N).
> >
> > This efficiency advantage stems from the fundamental difference in how each method handles computations. While our approach can easily run in parallel across multiple GPUs, parallelizing the Monte Carlo Tree Search (MCTS) component of LATS is still challenging. In fact, implementing parallelizable MCTS remains impractical using the original [LATS codebase](https://github.com/lapisrocks/LanguageAgentTreeSearch/blob/main/programming/run_lats_gpt4.sh), limiting its scalability and efficiency in PDDL domain synthesis.
> >
> > In MC5, we show that our is 44x faster than LATS.
> >
> > ## MC3: Planning Graph
> >
> > Thank you for raising the question about planning graph.
> >
> > Fig. 6 shows the planning graph of Termes, a very complex planning problem with 7 different actions. The planning graph is generated by this [repo](https://github.com/debbynirwan/planning_graph) using state-space searching. For detailed information on how the planning graphs are generated, reviewer can kindly refer to this [blog](https://towardsdatascience.com/improving-classical-ai-planning-complexity-with-planning-graph-c63d47f87018/).
> >
> > In Fig. 6, the first node shows the initial world state, it goes through several transformations by applying plausible actions until the solution plan is founded to reach the goal that declared in the PDDL problem code (i.e., the last node in the chain).
> >
> > We apologize that the graph was unclear, as the state-space is so big for Termes, which can reflect that it is very complex, we have made it a little bigger in the new version.
> >
> > ## MC4(a): Clarification on the Single Numbers in Table 2 and Figure 3
> >
> > As detailed in Section MC1, we employ the VAL validation toolkit as our evaluation metric for PDDL domains, as presented in Table 2 and Figure 3.
> > The notation BoN-8-Qwen2.5-Instruct indicates that validation is considered successful if any of the 8 generated solutions passes the VAL check.
> > For BoN configurations with N samples, we set K = min(N, 32), ensuring K does not exceed 32 regardless of N's value.
> >
> > In Section 4.6, the pass@K metric indicates whether at least one of the K plans generated by o1-mini matches the ground-truth plans.
> >
> > ## MC5: Compare with prompt-based baselines
> >
> > Thank you for highlighting the non-trivial prompt-based baselines.
> >
> > Directly re-implementing these baselines for **PDDL-domain synthesis** presents challenges due to three key factors:
> >
> > 1. **Design Mismatch**: Methods like LATS and RAP were not originally developed for PDDL-specific tasks, requiring substantial adaptation for our domain synthesis problem.
> > 2. **Reward Mechanism Complexity**: Both LATS and RAP are Monte Carlo Tree Search (MCTS) algorithms that depend on intermediate reward models. LATS generates Python test cases and uses their pass rates as rewards, while RAP combines diverse reward sources including action likelihood, task-specific heuristics, and self-evaluation by the language model.
> > 3. **Computational Limitations**: Neither LATS nor RAP supports any GPU parallelization. For our tasks, even with a maximum iteration number of four, LATS-MCTS required over three days to process all 283 PDDL instances.
> >
> > In our re-implementation of Reflexion, LATS, and RAP, we adopt a unified reward mechanism using the language model itself as the reward model, providing sparse binary rewards (correct/incorrect) for generated domains. This design choice rendered LATS and RAP functionally equivalent in our setup, as both rely on similar MCTS frameworks with different reward sources. We adopt the official repository of LATS to reimplement the method and get the results below, all the non-trivial methods are based on Qwen2.5-Coder 7B.
> >
> > | **Method** (NL2Domain) | VAL Pass Rate | Time |
> > | --- | --- | --- |
> > | [LATS-Reflextion](https://github.com/lapisrocks/LanguageAgentTreeSearch/blob/main/programming/reflexion.py) | 15.2 | 8 hours |
> > | [LATS-MCTS](https://github.com/lapisrocks/LanguageAgentTreeSearch/blob/main/programming/run_lats_gpt4.sh)  | 28.1 | 132 hours |
> > | iVML-5-BoN-8-Qwen2.5-Coder  | 85.2 | 3 hours |
> >
> > ## Minor Changes
> >
> > Thanks so much for the minor changes suggestions.
> >
> > 1. We have corrected the spacing issues and spelling errors identified by the reviewer.
> > 2. We have expanded the related work section to include relevant PDDL-related research.
> > 3. We have redrawed Fig. 2 with an enlarged format to enhance clarity and readability.

---

> > > ### Author Response · Authors · 2025-03-18
> > > **Response from the authors (Part 3)**
> > >
> > > ### Broader Impact
> > >
> > > We have incorporated a broader impact statement in Section 6, which discusses how our methods can advance the safe deployment of human-centered AI agents. Given that recent fully autonomous agents often produce unverifiable outputs and frequently generate hallucinations—including flawed goal achievement assessments, incorrect state transition estimations, and rule violations (see Figure 2)—our approach offers a critical advancement. By generating scalable symbolic world models, our methods produce verifiable, rule-followed plans that address these limitations, thereby supporting more reliable and trustworthy AI systems.
> > >
> > > For the human-in-the-loop experiment, we recruited one graduate student volunteer with a computer science background. The volunteer was compensated $13.80 USD for their participation in evaluating the ability to implement Termes code based on natural language descriptions of Termes.
> > > Since human-in-the-loop evaluation is not the primary research focus of this paper, we provide only limited observations for reference.
> > >
> > > ## References
> > >
> > > [1] C. Huang et al. On the Limit of Language Models as Planning Formalizers. In Arxiv, 2024
> > >
> > > [2] L. Guan et al. Leveraging Pre-trained Large Language Models to Construct and Utilize World Models for Model-based Task Planning. In NeurIPS, 2023
> > >
> > > [3] P. Smirnov et al. Generating consistent PDDL domains with Large Language Models  https://arxiv.org/pdf/2404.07751v1
> > >
> > > [4] Z. Max et al. Planetarium: A rigorous benchmark for translating text to structured planning languages arXiv:2407.03321, 2024. In Arxiv, 2024
> > >
> > > [5]  A. Zhou et al. Language Agent Tree Search Unifies Reasoning Acting and Planning in Language Models. In ICML, 2024.

---

> ### Comment · Reviewer_KXRw · 2025-03-25
>
> Thanks for the detailed response. Some contents in the response (e.g. experiment results) are not added to the paper yet, and hopefully the author could add these results to the paper as well. While the response addressed most of my concerns, one concern in my review remains:
>
> The model has potential to address the task very well (almost perfect as suggested by the total number of domains in Tab. 1), but for some reason its ability must be extracted via a very large number of reflections (essentially self-reflection as the backbone model for  $f_{\text{opt}}$ and $f_{\text{learner}}$ are the same). Could the authors give an explanation for such phenomenon?

---

> > ### Author Response · Authors · 2025-03-30
> > **Response from the authors**
> >
> > Great insight. We think this is exactly the point of test-time scaling, where we aim to find an effective way to take advantage of test-time compute. While Best-of-N sampling leverages test-time compute independently, iVML leverages test-time compute sequentially/conditionally. Therefore, the self-reflection style of iVML can be viewed as using the LLM itself to summarize and provide feedback for the previous reasoning tokens, making the sequential test-time scaling more effective. As an analogy, BoN resembles the idea of breadth-first search, while iVML is similar to the idea of depth-first search.

---

> > > ### Comment · Reviewer_KXRw · 2025-03-31
> > >
> > > Thanks for the further response. I do not have any other question and vote for acceptance.

---

### Review · Reviewer_AsXu · 2025-02-23

**Summary Of Contributions:**

* The authors propose a strategy to enable LLMs to plan. They claim that current LLMs still struggle with deductive reasoning and principled planning, so they aim to automatically generate PDDL domains either from natural language descriptions or from PDDL problems. However, this translation task remains challenging because of the scarcity of PDDL training data and the difficulty of maintaining logical consistency across predicates and actions.

* In more technical terms, the authors propose a strategy (depicted in Figure 1), based on using Best-of-N (BoN) sampling for PDDL domain initialization (that generates multiple chain-of-thought responses composed of PDDL domains and thoughts) and Instance Verbalized Machine Learning (iVML) to iteratively improve solutions (using a critic LLM that evaluates the previously generated solutions, and an LLM that learns froms this feedback and refines the PDDL domains, ie iteratively eliminates logical inconsistencies and grammatical errors). Within this framework, BoN emphasizes exploration while iVML emphasizes exploitation.

* The authors claim four contributions: (i) Scalable PDDL domain generation; (ii) Application of VML to instance learning; (iii) Efficient test-time compute scaling; and (iv) Robust planning through PDDL abstraction.

**Audience:**

Yes

**Broader Impact Concerns:**

No concerns regarding ethical implications.

**Claims And Evidence:**

Yes

**Requested Changes:**

My overall opinion of the paper is good and, therefore, none of the changes I propose below are very critical. However, if taken into account, they would make the paper stronger.

* Figure 1 includes an arrow from BoN Initialization to iVML. That arrow is labeled as $T_{0}$,$D_{0}$. However, if I'm not mistaken, it should be $T_i$, $D_i$ since not only Domain $D_0$ and Thought $T_0$ are passed to iVML. I assume that you optimize a set of initial solutions (generated by sampling diverse solutions (high temperature) and keeping the ones with highest log-likelihood), and not only the best of them (e.g. $D_0$), using the iVML strategy. Is this correct?

* Also, according to what is said in the text: "For each problem, the LLM generates N candidate solutions in parallel and retains the K samples with the highest log-likelihoods. [...] During the selection phase, the top K candidates with the highest scores are chosen as the initialization points to be optimized by iVML." In Figure 1, instead of "Thought $T_{N}$" I guess we should have "Thought $T_{K}$" (in other words, I think that we should use $K$ instead of $N$ in the figure).

* Is $f_{update}$ in Eq.4 (page 7) the same as $f_{learner}$ in Figure 1? If that's the case, please, standardize/unify terminology.

* It would be helpful to briefly describe what Verbalized Machine Learning (VML) is rather than assuming that all readers will be familiar with the concept. The same applies to single-pass sampling (page 11), whose difference with BoN is just very superficially described. Or even in the conclusions, where for the first and only time in the paper, and without any kind of explanation or clarification, the authors mention "the cold-start problem in formal language synthesis". In cases like this, it could be enough to mention that is challenging for LLMs to generate high-quality outputs in unfamiliar domains due to insufficient training data, and it often necessitates additional techniques to compensate for this initial lack of domain-specific experience.

* The related works section is acceptably good in my opinion, but it's not very clear to me what exactly are we seeing in Table 1. Apparently, it represents a summary and comparison with the current state-of-the-art in the field of PDDL domain generation. However, if I'm not mistaken, there are closely related works that are not included in there, and that are not mentioned in the paper whatsoever, like:
   * Smirnov, P., Joublin, F., Ceravola, A., & Gienger, M. (2024). Generating consistent PDDL domains with Large Language Models. arXiv preprint arXiv:2404.07751.
   * Oswald, J., Srinivas, K., Kokel, H., Lee, J., Katz, M., & Sohrabi, S. (2024). Large language models as planning domain generators. In Proceedings of the International Conference on Automated Planning and Scheduling (Vol. 34, pp. 423-431).
   * Mahdavi, S., Aoki, R., Tang, K., & Cao, Y. (2024). Leveraging environment interaction for automated pddl translation and planning with large language models. arXiv preprint arXiv:2407.12979.

In order to enrich the related works section, as well as to make the paper more complete and self-contained, my advice would be to introduce and discuss these references in the paper.

* Regarding the meaning and purpose of thoughts ($T_i$):
   * Are you employing the long CoT internally used, for instance, by OpenAI's reasoning models (that are hidden from the end-user when interacting with the model) or what the user sees as a summary of the long CoT (concise and optimized for human-readability)? In this sense, could you give more details about how do you extract $T_i$ from the output of the LLM in order to provide it to the $f_{opt}$ and $f_{update}$ functions/LLMs?
   * In Eq. 4, what is the reason behind using the thoughts ($T_i$)? Why isn't using just the feedback $F_i$ enough to iteratively improve the generated domain?

* BoN generates a series of initial solutions (domains, thoughts) that are sent to iVML for iterative refinement. Once there, through a series of iterations, these solutions are improved. But, once the process is finished, how is the best generated domain chosen to be sent to the planner? Is the same criterion of the previous step used again: "Each candidate $c_i$ is assigned a score $S_i$ based on the sum of the log-likelihoods of its generated tokens"?

* In page 8, the authors state that "The generated PDDL-based world model, combined with a classical planner in the loop, helps to reduce hallucinations when using LLMs directly as planners." Where is this analyzed, evaluated and measured in the paper? I would try to be cautious when making statements. If the reduction in hallucinations is not evaluated or quantified, I would pose it as a mere hypothesis or as a possible future work to consider.

* Regarding Table 2:
   * It's a bit hard to interpret. I would recommend to highlight the best results in bold letters. Furthermore, if one wants to compare the relative improvement obtained from, for instance, Qwen2.5-Instruct 72B to BoN-8-Qwen2.5-Instruct 72B and, from there, to iVML-5-BoN-8-Qwen2.5-Instruct 72B, it's not quite straightforward/immediate. I would try to visualize or structure the information so that the subsequent analysis, carried out in section 4.2, is more direct and simple for the reader.
   * The authors compare 5 LLMs with different number of parameters with different versions of Qwen2.5 augmented with BoN and/or iVML. Why did they not apply BoN+iVML to Llama3.1, Yi-1.5 and Yi as well?

* In the experimental part, different LLMs are compared and different ablation studies performed, but where is the comparison with other PDDL domain generation methods? In this sense, when in page 7 is written "[we] compare our test-time scaling algorithm to existing state-of-the-art methods" I guess it would be more accurate to say "compare our test-time scaling algorithm to the domain generation capabilities of different state-of-the-art LLMs". In any case, of course, it would be really great to include experiments comparing the performance of this framework with some other PDDL domain generator.

* In page 10, authors state that "these methods depend on two parameters: the sampling budget for BoN N, and the number of training epochs for iVML T." What about K in BoN? Remember that "For each problem, the LLM generates N candidate solutions in parallel and retains the K samples with the highest log-likelihoods. [...] During the selection phase, the top K candidates with the highest scores are chosen as the initialization points to be optimized by iVML." So, I assume that these K solutions are the ones that go to the next phase (iVML). How K was selected, and what is its impact in performance? In the same line of thought, why the authors selected $T=0.7$? What kind of experiment performed to conclude that this $T$ value was the most suitable one?

* Regarding Figure 3:
   * why use absolute values and not percentages, as has been done until now?
   * It's a bit confusing to use variable T in the paper to represent two different concepts: temperature (on page 9) and the number of iVML training epochs (on page 10).
   * I don't fully get why, based on the results displayed in Figure 3, the main results showed in Table 2 are obtained employing $T=5$ and $N=8$. I mean, why $T=5$ and $N=8$ were the chosen values? Could the authors comment on that?

* Sections 4.5 and 4.6 are a bit confusing as well. From the very beginning of the paper, we have been discussing and working on PDDL domain generation, but now we suddenly move to PDDL problem generation (Section 4.5), which is a different task:
   * How does the proposed framework change with this new task? If I'm not wrong the authors are trying to solve now an NL2Problem ("generate precise PDDL problems from natural language descriptions"). Are you providing as input the PDDL domain file as well? If the answer is no, does it mean that the model has to synthetically generate both (PDDL domain and problem files)?
   * Why again there are no comparisons (either experimental or methodological/conceptual) with current state-of-the-art PDDL problem generation approaches, like Núñez-Molina, C., Mesejo, P., & Fernández-Olivares, J. (2024). Nesig: A neuro-symbolic method for learning to generate planning problems. In ECAI 2024 (pp. 4084-4091)?
   * I understand that Table 4 shows the percentage of generated PDDL problems that are solvable by a planner. Which planner is used to calculate the Success Rate: Fast Downward as in Table 5? Where do the PDDL domain files that are entered as input to the planner come from (are they, in all cases, generated by the same models that generate the PDDL problems, or all the methods share the same PDDL domain file)? How exactly are the competitor LLMs finetuned? Some more information would be beneficial to better understand the work done in this section.
   * Regarding Section 4.6:
      * The proposed method (called "PDDL as Abstraction" in this section) is compared with LLMs that are used directly as planners (called "LLM-as-Planner Methods"). I assume that, in the case of "LLM-as-Planner Methods", they are given as input a description of the domain (in natural language) and the PDDL problem, and they should directly produce a plan. And in the case of "PDDL as Abstraction", a description of the domain (in natural language) is given as input and, after applying iVML, the generated PDDL domain and a PDDL problem are given as input to the planner (Fast Downward) in order to generate a plan. Is this correct?
      * What is Plan Accuracy? Is it the same metric as Success Rate (defined in page 8)?
      * The term "PDDL as Abstraction" I think would need further explanation, contextualization and/or justification.

* Regarding the contributions of the work, I strongly agree with the approach provided by Andrej Karpathy in https://karpathy.github.io/2016/09/07/phd/: _"Identify the core contribution. Before you start writing anything it’s important to identify the single core contribution that your paper makes to the field. I would especially highlight the word single. A paper is not a random collection of some experiments you ran that you report on. The paper sells a single thing that was not obvious or present before. You have to argue that the thing is important, that it hasn’t been done before, and then you support its merit experimentally in controlled experiments. The entire paper is organized around this core contribution with surgical precision. In particular it doesn’t have any additional fluff and it doesn’t try to pack anything else on a side. As a concrete example, I made a mistake in one of my earlier papers on video classification where I tried to pack in two contributions: 1) a set of architectural layouts for video convnets and an unrelated 2) multi-resolution architecture which gave small improvements. I added it because I reasoned first that maybe someone could find it interesting and follow up on it later and second because I thought that contributions in a paper are additive: two contributions are better than one. Unfortunately, this is false and very wrong. The second contribution was minor/dubious and it diluted the paper, it was distracting, and no one cared. "_. I mention this because, after having finished reading the paper, I'm not sure that the 4 contributions indicated by the authors are at the same level and have the same weight in the paper. My advice to the authors would be to be more precise at identifying their key contribution and highlight it more clearly.

* Typos:
   * page 5: "before LLM iteratively the internal logic with iVML" (I think a verb is missing here).
   * page 7: "the best-of-N sampleing" --> "the best-of-N sampling"
   * In Table 2, I guess the "Avg. (%)" column corresponds to the average of "NL2Domain (%)" and "Problem2Domain (%)". If that's the case, there is a typo since for the third row (Qwen2.5-Instruct with 3B params) you have that the average of "2.1" and "1.5" is 0%.
   * Conclusions: "OpenAI’s o1-mini.Our hybrid" A space is missing between "mini" and "Our".

**Strengths And Weaknesses:**

Strengths:
* The authors propose a framework to enhance LLM planning capabilities without using any training data (no finetuning).
* The proposed method is simple and effective.
* At an empirical level, the results obtained by the method are, in my opinion, very promising. As can be seen in Table 2, a significant improvement in performance is obtained with only 5 iterations of iVML.
* The paper is overall clear, well-written, well-presented and, as a consequence, easy to follow.
* I consider that improving the deductive reasoning and planning capabilities of LLMs is a  very pertinent and relevant research direction.

Weaknesses:
* In my humble opinion, some parts of the paper are a bit confusing. For instance, when at the beginning of section 3 the authors say that one of the two key questions they try to explore with their work is "How can PDDL serve as a good world representation for planning?" (question that they answer/develop in section 3.1). I think this question conveys the impression that the paper is interested in the theoretical foundations of why PDDL is good for automated planning when, in fact, that is not the focus of the paper at all.
* Related to the previous point, I think there are some aspects of the paper that could/should be clarified (see my feedback below in the "Requested Changes" section of this review).
* Although the paper focuses on PDDL domain generation, their proposal is not experimentally compared with other PDDL domain generation methods.

---

> ### Author Response · Authors · 2025-03-18
> **Response from the authors (Part 1)**
>
> We want to sincerely thank the reviewer for the constructive and thorough comments and suggestions on our work. We take every comment seriously and hope our response can address the reviewer’s concerns. If there are any remaining questions, we are more than happy to address them. In the following responses, we use RC to represent requested changes.
>
> ---
>
> ## RC1, 2, 3: Notations in Fig 1
>
> Thanks for the suggestion. We have re-plotted Fig. 1 and use the correct and consistent notations as the reviewer suggested.
>
> ## RC4: Concepts Explanation
>
> We have added the detailed concept explanation of VML (i.e., in Sec. 2.3), single-pass sampling (i.e., in Sec. 4.4), and cold-start (i.e., in Sec. 5).
>
> ## RC5: Add and Compare with existing PDDL-related works
>
> Thanks so much for your suggestions on the PDDL-related works.
>
> We have incorporated a discussion regarding the distinctions between our methodology and those presented in [1][2][3] within Section 2.2.
>
> Generally, these methods employ iterative refinement processes for generated PDDL domains. These approaches access oracle environments—consisting of ground-truth PDDL problems and PDDL domains—and validate the generated PDDL domain by the feedback that compares the resulting plans against ground-truth plans generated by the oracles.
>
> By contrast, our approach generates PDDL domains in an oracle-agnostic manner, focusing on scalability to scenarios where rule-based ground-truth signals are unavailable.
>
> In the following experiments, we compare our methods with oracle interactions method (e.g. [3]) on the tasks they have [pre-built oracle environments](https://github.com/BorealisAI/llm-pddl-planning/tree/main/data/domains). We use the planning accuracy metric: for a given planning problem (e.g., Termes), it measures the percentage of generated plans from the synthesized domains that exactly match the ground-truth reference plans.
>
> |  | Oracle Interaction [3] | iVML-5-BoN-4-Qwen2.5-Coder | iVML-5-BoN-8-Qwen2.5-Coder |
> | --- | --- | --- | --- |
> | Grippers | 0.0 | 100.0 | 100.0 |
> | BlocksWorld | 100.0 | 71.4 | 81.0 |
> | Termes | 100.0 | 100.0 | 100.0 |
> | Floortile | 0.0 | 0.0 | 100.0 |
>
> We found that our method is comparable to the oracle interaction method on the cases they considered.
>
> ## RC6: About the Thought T_{i}
>
> Current state-of-the-art LLMs have been extensively trained to generate lengthy Chain-of-Thought (CoT) explanations during both the supervised fine-tuning (SFT) and reinforcement learning (RL) stages [4].
>
> Our approach does not incorporate additional complex CoT mechanisms (e.g., hidden-CoT, self-reflection, etc.)  For current models, users typically need not explicitly prompt LLMs for complex CoT, as these models naturally combine natural language explanations with code during generation. we show our prompt clearly in Appendix D.
>
> We filter the code using standard regular expressions to identify and extract content between the '\```pddl' and '\```' pattern and extract the contents in between.
>
> The motivation for introducing T_{i} stems from the limitations of current LLMs in processing symbolic languages. By incorporating T_{i}, we enable LLMs to interpret symbols within natural language contexts, thereby enhancing the interpretability of the refinement process.
>
> ## RC7: Hallucination
>
> In our context, the hallucination refers to the rule violation, incorrect state transition estimation, incorrect goal achievement estimation, and incorrect self-evaluation during LLMs’ planning with natural language (See Sec. 4.6). We exhibit a demonstration of o1-mini hallucinates when dealing with Termes (i.e., Fig. 2), and thats why in Table 5, o1-mini only has 52.4 accuracy on Termes.
>
> PDDL-based planning is way different from natural language, it uses state-based planning to plan. During planning, PDDL is able to model the explicit state-transition to avoid incorrect state transition estimation, rule violation, etc. When combined with A* solver, it is able to guarantee optimality and satisfaction for classical planning problems, which enhances the trustworthiness of its plan.
>
> ## RC8: The selection of best generated domain
>
> The best domains for planning are selected by two criteria: 1. passing the PDDL validation tool VAL and 2. achieving the highest $S_{i}$ among all the candidates.
>
> ## RC9: Add Yi and LLaMa in Fig.2
>
> We have revised Figure 2 to highlight the best results and incorporated the results of the test-time scaling of Yi and LLaMa.

---

> ### Author Response · Authors · 2025-03-18
> **Response from the authors (Part 2)**
>
> ## RC10: The hyperparameters in iVML
>
> For BoN configurations with N samples, we set K = min(N, 32), ensuring K does not exceed 32 regardless of N's value.
>
> For most of the experiments in Sec. 4, main experiments, we adopt K=8, and T=5.
>
> This configuration represents a reasonable computational affordance, particularly when considering our experimental setup utilizing 8 H100 GPUs.
>
> ## RC11: The issue in Fig. 3
>
> In Figure 3, we converted the exact values to percentages, and we change to use $\tau$ to represent the temperature.
>
> ## RC12: PDDL Problem Generation
>
> We investigate natural language to PDDL problem synthesis because we want to check if the iVML’s advantage on PDDL domain synthesis can also benefit other symbolic code generation tasks in PDDL language (e.g., PDDL problem synthesis), which would demonstrate its
> generalization abilities.
>
> While our paper primarily focuses on PDDL domain synthesis and NL2Problem is a distinct task, both fall under the broader category of *natural language-to-symbolic world model synthesis*. This shared objective justifies their comparison.
>
> Importantly, during PDDL problem generation, no pre-existing PDDL domain is provided as input—only natural language descriptions guide the synthesis process.
>
> We exclude comparisons to [5] as our work centers on generating symbolic world models directly from natural language, whereas [5] requires a PDDL domain as input and does not process general natural language instructions. This fundamental difference in input modalities makes direct comparison inappropriate.
>
> ## RC13: Regarding Section 4.6
>
> In Section 4.6, we employ the Fastdownward system (utilizing the A* algorithm) to evaluate our approach.
>
> In Sec. 4.6, We redefine the term PDDL-as-abstraction as PDDL as Planning Representation to avoid misunderstanding. This refers to our methods that generate a PDDL domain as a symbolic world model, paired with a corresponding PDDL problem instance, and applying A* search to generate plans.
>
> For the LLM-as-Planner approach, both the domain and problem are represented in natural language. For reference, see [this repository](https://github.com/Cranial-XIX/llm-pddl/tree/main/domains/termes), which provides planning problems for the Termes domain in both symbolic (PDDL) and natural language formats with equivalent semantics.
>
> Planning accuracy is measured as the percentage of generated plans from synthesized domains that exactly match the ground-truth reference plans (see Section 4.6).
>
> ## RC13: Clarify the Contributions of Our Works
>
> Thanks so much for your suggestions on how to clarify the core research contributions.
>
> We have rewrote the content in beginning of Sec.3 to better exhibit the work’s core contribution. The central research question we focus is the effective generation of symbolic world models through test-time scaling approaches.
>
> ## RC14: Typos
>
> Thanks so much for pointing out the typos. We have fixed every typos the reviewer mentioned.
>
> ## References
>
> [1] Smirnov, P., Joublin, F., Ceravola, A., & Gienger, M. (2024). Generating consistent PDDL domains with Large Language Models. arXiv preprint arXiv:2404.07751.
>
> [2] Oswald, J., Srinivas, K., Kokel, H., Lee, J., Katz, M., & Sohrabi, S. (2024). Large language models as planning domain generators. In Proceedings of the International Conference on Automated Planning and Scheduling (Vol. 34, pp. 423-431).
>
> [3] Mahdavi, S., Aoki, R., Tang, K., & Cao, Y. (2024). Leveraging environment interaction for automated pddl translation and planning with large language models. arXiv preprint arXiv:2407.12979.
>
> [4] Yu, P., Xu, J., Weston, J., & Kulikov, I. (2024). Distilling system 2 into system 1. *arXiv preprint arXiv:2407.06023*.
>
> [5] Núñez-Molina, C., Mesejo, P., & Fernández-Olivares, J. (2024). Nesig: A neuro-symbolic method for learning to generate planning problems. In *ECAI 2024* (pp. 4084-4091). IOS Press.

---

> > ### Comment · Reviewer_AsXu · 2025-03-21
> >
> > I thank the authors for taking into consideration my and other reviewers' comments into consideration. I appreciate all the additional explanations and responses provided.
> >
> > In any case, there are minor aspects that, in my opinion, could/should be addressed:
> >
> > - caption Figure 1:
> >    - it still mentions $f_{learner}$, whereas, if I'm not mistaken, this expression is no longer used in the paper. It should be $f_{update}$.
> >    - I believe it would be helpful to clarify in the caption what the indices ($N$, $k$, $i$) represent.
> >
> > - Several aspects are, in my opinion, well addressed by the authors in their OpenReview response but, again if I'm not wrong, are not included in the revised version of the paper. My recommendation would be to enrich the paper with this information so that it's more self-contained and a potential reader can fully understand all the details. For instance:
> >    - The authors mention that _"We found that our method is comparable to the oracle interaction method on the cases they considered."_ Are they planning to introduce a table or a summary of the main results in the paper?
> >    - "RC12: PDDL Problem Generation": _"Importantly, during PDDL problem generation, no pre-existing PDDL domain is provided as input—only natural language descriptions guide the synthesis process."_ I suggest including this information explicitly in the paper to make this point clearer and avoid misunderstandings.
> >
> > Apart from these details, I'm happy with the paper (in terms of form, content, technical soundness, and clarity of the narrative and argument). I want to congratulate the authors and reviewers on their work.

---

> > > ### Author Response · Authors · 2025-03-30
> > > **Response from the authors**
> > >
> > > Thank you so much for the follow-up suggestions. We have revise the revision paper as the reviewer requested:
> > >
> > > ## Minor1: Caption Figure 1
> > > 1. We have corrected the mistake of $f_{update}$ and $f_{learner}$.
> > > 2. We have clarify the caption of $(N, k, I)$ as follow:
> > > *Here, $N$ represents the total number of candidate solutions generated, $k$ is the index of the top candidates retained for further optimization (with $K$ indicating the total number of such candidates), and the index $i$ is used to denote the iteration step within the optimization procedure.*
> > >
> > > ## Minor2: Add the Comparison Results in the Paper
> > > In Table 5, we supplement the results by comparing our method with the oracle interaction method. Additionally, in Section 4.6, we provide a detailed analysis of these comparisons as follows:
> > >
> > > *We compare our method with a PDDL-based approach~\cite{mahdavi2024leveraging} that uses the oracle environment to refine the answers.
> > > By contrast, our approach generates PDDL domains in an oracle-agnostic manner, leveraging test-time scaling to enhance scalability in scenarios without rule-based ground-truth signals. We found that our method (\eg, BoN-4-iVML-5) achieves comparable performance to the oracle interaction method in the cases they considered, with both achieving 100.0 on Termes. However, our method outperforms the oracle interaction method on Floortile and Grippers.*
> > >
> > > Also, in Section 4.5, we have included the information *Importantly, during PDDL problem generation, no pre-existing PDDL domain is provided as input—only natural language descriptions guide the synthesis process.*.

---

> > > > ### Comment · Reviewer_AsXu · 2025-03-30
> > > >
> > > > I thank the authors for their second round of responses. I have no further comments.

---

### Review · Reviewer_grBB · 2025-03-04

**Summary Of Contributions:**

Large language models (LLMs) struggle to solve complex planning problems with
sequential reasoning because the innate ambiguity in natural language results in
constraint violations and incorrect world modelling. This paper proposes a
methodology to allow LLMs to generate Planning Domain Definition Languages
(PDDLs), which can be used to create symbolic world models that allow classical
planning algorithms to then be used to solve planning problems. Their method
allows for test-time data generation of training data for PDDL generation. This
approach combines Best-of-N sampling to generate a candidate set of solutions
for PDDL generation with instance verbalized machine learning to refine the
solutions. They then evaluate their approach on a suite of experiments in a
number of PDDL generation tasks. They show that their approach helps LLMs
solve complex planning problems in a variety of domains.

**Audience:**

Yes

**Broader Impact Concerns:**

I have no concerns.

**Claims And Evidence:**

Yes

**Requested Changes:**

- Page 5, last paragraph: "... before LLM iteratively the internal logic with
iVML, ...". The sentence makes no sense.
- Page 7, equations: Clarify that $f_{opt}$ and $f_{update}$ are the NL
parameterized functions.
- Section 3.3: This should be explained generally. I am having difficulty
following the iVML framework. Maybe explain what the functions take as input and
returns as output. Generally define $\mathcal{G}$. Define $\mathbf{T}$.
- Section 4.3: "BoN sampling undergoes two phases." is confusing --- sounds as
if the algorithm has two phases. And what does "improve sublinearly" mean?
- BoN vs Single-pass sampling: I am not sure what the section is getting at. Is
this equivalent to saying taking more than one sample is better? That seems
like an obvious conclusion. What exactly is "Single-pass sampling"?
- Figure 1: Explaination -> Explanation
- Figure 1: The paper does not refer to the figure at any point nor does it
actually help in understanding the proposed algorithm. Maybe get rid of it and
use the space to add to other sections or use the figure to further improve the
explanation in Section 3?
- Explain the temperature parameter or specifically where it is used in sampling
  e.g., add an equation.
- Figure 3: Swap the diagrams. As you explain BoN results first, use that as the
  first diagram.
- Section 4 may need a rewrite. It was not easy at all to get through. While I
appreciate the thorough experimentationn, I found myself getting lost at the
objcectives. While there is some organization currently, the lack of explicitness made it hard to see or remember the purpose of the experiments. Some thoughts on re-organizing the section:
    - Perhaps organize the experiments by claim. For example, "Claim: BoN provides
    better initialization for iVML" then follow with the experiments, results
    and how the results back up the claim.
    - Divide the experiments by type/domain.

**Strengths And Weaknesses:**

**Strengths**
- The strength of the approach proposed by the authors is that it is
conceptually simple and it seems easy to implement in practice.
- The authors also performed a swath of experiments to provide empirical
proofs of their performance claims. Most of the claims made in the paper are
supported by the results.
- Overall I found the paper interesting, with a decent motivation and a reasonable approach to address a problem.

**Weaknesses**

There are a few issues of note:
- On the bottom of page 12: The claim that their method
"prevents catastrophic forgetting" is not substantiated by the results in Table
4 or at least I fail to see how it does. One could argue that it may not suffer
from it but that is not the same thing is preventing it.
- Page 11, BoN vs Single-pass sampling: I do not see the purpose of this set of
  experiments. There is not an adequate description of single-pass sampling for
  me to understand why it is a baseline. If it is equivalent to BoN with an
$N=1$ then all this paragraph underscores is that more samples are better?
- Page 11, "BoN generates a diverse set of initial candidate solutions": How is
  this claim backed up? The results show that BoN helps but not that the
solutions generated are diverse.
- The experiments and results section was actually difficult to go through and a
  little unclear.

---

> ### Author Response · Authors · 2025-03-18
> **Response from the authors (Part 1)**
>
> We want to sincerely thank the reviewer for the constructive and thorough comments and suggestions on our work. We take every comment seriously and hope our response can address the reviewer’s concerns. If there are any remaining questions, we are more than happy to address them. In the following responses we use RC to represent requested changes.
>
> ---
>
> ## RC 1, 2, 6, 7, 9: Minor Changes
>
> Thank you for raising these important questions.
>
> **RC1:** In Sec. 3.2, we change the sentence ”before LLM iteratively the internal logic with iVML” to “the second stage is to iteratively refine the initial solutions via an instance verbalized machine learning algorithm”.
>
> **RC2**: In Sec. 3.3, we have clarify that “in iVML, functions are parameterized using natural language”.
>
> **RC6 and RC7**: In Fig.1, we have corrected the spelling of “Explanation” and added a note: “See Figure 1 for the overview of our method in Sec. 3.”
>
> **RC9**: We have swapped the diagrams as suggested.
>
> ## RC3: Define $\mathcal{G}$ and $\textbf{T}$
>
> In section 3.3, we have clearly clarified the definition of $\mathcal{G}$ and $\textbf{T}$.
>
> For  $\mathcal{G}$:  “Here,   $\mathcal{G}$ is always within the language spaces for all the tasks we considered, for example, in NL2Domain task, $\mathcal{G}$ corresponds to a natural language description of the planning domain, for Prob2Domain, $\mathcal{G}$ represents a PDDL problem instance.”
>
> For  $\textbf{T}$: “Here,  $\textbf{T}$  is generated by applying chain-of-thought (CoT) sampling at each step of the PDDL domain generation process, it comprises all non-PDDL components in the outputs of $f_\mathrm{update}(\cdot)$ excluding the final structured PDDL domain itself.”
>
> ## RC 4: Address the Confusions of BoN sampling
>
> Thank you so much for raising the questions.
> In Section 4.3, we have rewrite the sentence “BoN sampling undergoes two phases” to “we observe two distinct phases in the phases in the performance of BoN sampling as the sampling budget N increase”.
> Here the term “phases” refers to the two distinct pattern when increasing the sampling budget N. phase1:  Both Prob2Domain and NL2Domain demonstrate a steep increase in the percentage of domains passing validation as the sampling budget increase, and phase2: Performance reaches saturation with observable degradation trends. It does not means the the algorithm has two procedures to work.
>
> ## RC 5: BoN vs Single-pass sampling and Weakness 2
>
> Section 4.4 studies the sensitivity of iVML to solution initialization and whether a diverse set of
> candidates can enhance the performance of iVML. And we have clarified the difference between single-pass sampling and BoN in Sec. 4.4.
>
> “Single-pass sampling by requesting the LLMs to generate only once as the solution initialization for the iVML process, and 2. BoN sampling by requesting the LLMs to generate diverse candidate solutions for initializing the optimization process using high-temperature sampling.”
>
> ## RC8: Temperature Equation and Weaknesses 3
>
> We have added the equation for temperature sampling in Section 3.2, Equation 3.1.
>
> The purpose of having temperature is to adjust the sampling distribution.
> Higher temperature value (e.g., $\tau\geq0.7$) increases randomness and diversity in the generated candidates by flattening the probability distribution, giving relatively more weight to lower-probability tokens, in contrast, a lower temperature (e.g., $\tau\leq0.3$) sharpens the distribution, favoring high-probability tokens and producing more deterministic outputs.
>
> And this explains the reviewers concern that “The results show that BoN helps but not that the solutions generated are diverse.” as we use high-temperature sampling to make sure the diversity in single response, by sampling more times, we can get a candidates set with more diversity.

---

> > ### Author Response · Authors · 2025-03-18
> > **Response from the authors (Part 2)**
> >
> > ## RC10: Rewrite Sec. 4
> >
> > Thank you for your suggestions on rewriting section 4.
> > At the beginning of Section 4, we have clarified the main research question we want to investigate in each subsection.
> >
> > - Section 4.2 examines state-of-the-art LLMs' capabilities on PDDL domain synthesis and evaluates whether test-time scaling approaches enhance PDDL generation quality.
> > - Section 4.3 analyzes the convergence patterns of BoN and iVML, highlighting the specific improvements iVML offers over BoN.
> > - Section 4.4 investigates iVML's sensitivity to initialization conditions and whether diverse candidate sets can boost iVML performance.
> > - Section 4.5 explores whether iVML's advantages in PDDL domain synthesis extend to other symbolic code generation tasks within the PDDL language (e.g., PDDL problem synthesis), demonstrating its broader applicability.
> > - Section 4.6 assesses our planning approach (generating PDDL domains via iVML followed by heuristic search-based planning) against LLM-based planners on complex planning problems.
> >
> > ## Weakness 1:  Catastrophic forgetting
> >
> > Thank you for raising the concern. As one of the test-time compute scaling methods, our algorithm is training-free and does not need to finetune the LLM, unlike [1] where SFT is performed with more than 130k samples of PDDL problems. Because our algorithm is performed purely in-context without the need to modify the weights of LLMs, it will not suffer from catastrophic forgetting of LLM’s pre-trained knowledge (eg, LLM’s general instruction ability and chain-of-thought reasoning).
> >
> > However, we agree with the reviewer that this statement can be revised to be more accurate. We revise the original statement to “better preservation of pre-trained knowledge”.
> >
> > # References
> >
> > [1] Max Zuo, Francisco Piedrahita Velez, Xiaochen Li, Michael L Littman, and Stephen H Bach. Planetarium: A rigorous benchmark for translating text to structured planning languages. arXiv preprint arXiv:2407.03321, 202

---

> > > ### Comment · Reviewer_grBB · 2025-04-04
> > > **Acknowledgment**
> > >
> > > Thank you for the responses and the changes to the paper.
> > >
> > > I vote to accept.

---

### Review · Reviewer_CeE4 · 2025-03-12

**Summary Of Contributions:**

This paper presents a novel method for generating symbolic world models (in the form of PDDL planning domains) from natural language descriptions, using a combination of two forms of a test-time compute scaling (best-of-N sampling and iterative refinement between an LLM critic and an LLM improver). The method is compared with several other LLM methods, and is shown to improve planning performance via the application of search to the generated domains.

**Audience:**

Yes

**Broader Impact Concerns:**

There are no concerns about ethical implications of this work.

**Claims And Evidence:**

Yes

**Requested Changes:**

Please see the questions and suggestions in the section above.

**Strengths And Weaknesses:**

This paper presents a very creative solution combining LLM-driven synthesis of symbolic world models with test-time compute. Although the full method is somewhat complicated, all of the components are well-motivated, and the experimental results clearly demonstrate significant improvements over several other competitive approaches. The evaluation and experiments are generally very thorough. There is a very informative evaluation of the tradeoffs and convergence properties of the two test-time scaling techniques (BoN and iVML), and a thorough comparison several LLM baselines and fine-tuned models. I have a few questions and suggestions below, but overall I think this paper makes a great contribution.

## Questions and suggestions:
- In the introduction, there are several statements that should be accompanied by citations, e.g. claims about the inadequacy of LLMs and o1 on planning problems.
- It was not entirely clear to me what metric is presented in Table 2. Is this a measure of the accuracy of the generated PDDL domains (vs. the accuracy of PDDL domains generated by LLMs), or a measure of the accuracy of plans generated by using these domains (vs. the accuracy of plans generated directly by LLMs)? Do these problems have a ground-truth PDDL that can be directly compared with the PDDL generated by the proposed approach, and, if not, can such a comparison be performed?
- I appreciated the analysis of convergence for BoN and iVML, and the characterization of their benefits in terms of exploration vs. exploitation. However, it wasn't clear to me whether the extra test-time compute introduced by iVML is controlled for. In other words, when comparing a model that uses BoN with a model that uses BoN + iVML, the iVML introduces additional test-time computation. It seems appropriate to have a control condition where the amount of computation performed is balanced between the BoN model and the BoN + iVML model (i.e. there are more samples drawn from the BoN-only model, in proportion to the additional test-time compute used by the addition of iVML).

---

> ### Author Response · Authors · 2025-03-18
> **Response from the authors**
>
> We want to sincerely thank the reviewer for the constructive and thorough comments and suggestions on our work. We take every comment seriously and hope our response can address the reviewer’s concerns. If there are any remaining questions, we are more than happy to address them. In the following responses, we use QS to represent Questions and suggestions.
>
> ---
>
> ## QS1:  Statements accompanied by citations
>
> Thanks for your suggestions.
> We have added more citations in the introduction.
>
> For example, [1] show that o1 remains challenging in complex planning scenarios like Barman, Floortile, and Termes. [2] show that LLMs still struggle with deductive reasoning and principled planning, etc.
>
> ## QS2:  Metric in our paper
>
> Thank you for raising these important questions. We clarify our evaluation metrics as follows:
>
> 1. Domain Synthesis Accuracy (Table 2 and Fig. 3)
>
>     A synthesized domain is considered “accurate” if it passes all syntax and semantic validation checks using the [VAL](https://github.com/KCL-Planning/VAL) toolchain, following the previous works [1][2][3]. VAL is able to check the syntax errors in the pddl domains e.g. invalid predicates parameters types, invalid predicate name, invalid predicate format, incorrect predicate usage, etc [3].
>
>     Our accuracy metric represents the percentage of domains that fully pass VAL validation.
>
>     Note that even passing the VAL checking is not a simple task, (i.e. Llama3.1-Instruct-70B only achieves 1.1% and 0.0% VAL passing rate on NL2Domain and Prob2Domain tasks).
>
>     We have change the “accurate" in Fig. 3 to the “#Domains pass VAL”, and clarify the proper the clear definition of the metric used in PDDL domains generation in the Sec. 4.1.
>
> 2. PDDL problem success rate (Tab. 4)
>
>     Following the original benchmark Planetarium, we use the percentage of correct generated problems (i.e., the generated symbolic goals match with the GT goals). We have clearly clarified the metric used in PDDL problem generation in Sec. 4.5.
>
> 3. Planning Accuracy (Tab. 5)
>
>     For a given planning problem (e.g., Termes), it measures the percentage of generated plans from the synthesized domains that exactly match the ground-truth reference plans. See Sec. 4.6.
>
>
> ## QS3:
>
> In the analysis of convergence for BoN and iVML, we aimed to investigate the characteristics and limitations of BoN in PDDL domains synthesis, rather than strictly controlling for computational equivalence. Specifically, our focus was on understanding how iVML enhances the initial solutions provided by BoN, especially in scenarios where BoN alone may struggle.
>
> In this section, we aim to highlight the following :
>
> 1. We observed that BoN tends to saturate in terms of performance around N=32, as illustrated in Figure 3. This suggests that increasing N beyond this point does not yield significant improvements in the context of exploration. For particularly challenging tasks, even with a larger N (e.g., N=256), BoN alone may not provide adequate solutions, underscoring the necessity of using iVML for refinement.
> 2. In table 3, we observed that iVML leverages an in-context self-refinement mechanism
> to (1) identify constraint violations (e.g., illegal block stacking or invalid preconditions); (2) generate counterfactual natural language feedback to guide revisions; and (3) strengthen domain-specific reasoning priors through iterative updates.
>
> ## References
>
> [1] Kevin Wang, Junbo Li, Neel Pratik Bhatt, Yihan Xi, Qiang Liu, Ufuk Topcu, and Zhangyang
> Wang. On the planning abilities of OpenAI’s o1 models: Feasibility, optimality, and generaliz-
> ability. ArXiv, abs/2409.19924, 2024
>
> [2] Kewei Cheng, Jingfeng Yang, Haoming Jiang, Zhengyang Wang, Binxuan Huang, Ruirui
> Li, Shiyang Li, Zheng Li, Yifan Gao, Xian Li, et al. Inductive or deductive? rethinking the
> fundamental reasoning abilities of llms. arXiv preprint arXiv:2408.00114, 2024.

---

> > ### Comment · Reviewer_CeE4 · 2025-03-19
> >
> > Thank you for these responses. I have a few followup comments and questions:
> > - The reference cited here re: o1's planning abilities does not appear to be referenced in the introduction when this claim is made.
> > - The claim refers to o1, but the paper investigates o1-preview.
> > - In the related work, there is discussion of the limitations that LLMs face in planning problems, but this seems to be in reference both to standard LLMs and reasoning models like o1. These two types of models have a very different set of capabilities with respect to planning, so I think it is better to distinguish between them when discussing previous work.
> > - Thank you for the clarifications regarding the metrics used. I think it would be helpful to include this clarification at the beginning of section 4.1 (in 'Evaluation tasks and datasets') and also provide some explanation in the caption to Table 2.
> > - I appreciate the additional explanation regarding limitations of BoN. Given that you have investigated how the performance of BoN alone scales with the number of samples, shouldn't it be straightforward to establish that the added benefit of iVML goes beyond the benefit of using the same amount of additional compute for generating more samples using BoN?

---

> > > ### Author Response · Authors · 2025-03-30
> > > **Response from the authors**
> > >
> > > Thanks so much for the follow-up questions.
> > >
> > > ## QS1: o1 Planning Citation in Introduction
> > >
> > > We used to follow the previous works[2][3] by using the term “o1” to represent the o1-series models (e.g. o1-mini, o1-preview).
> > >
> > > We have revised the introduction to explicitly cite the planning capabilities of the specific "o1-series" models (o1-preview and o1-mini). The revised text now reads: "*While recent LLMs like OpenAI-o1-mini and OpenAI-o1-preview, ....,  they still struggle with deductive reasoning and principled planning [1, 2] that requires the consideration of optimality, constraints, and
> > > complex state transitions.*"
> > >
> > > To address the reviewer’s suggestion, we have replaced the generic term "o1" with the concrete model names "o1-preview" and "o1-mini" in all citations and throughout the manuscript. This ensures clarity and precision when referencing these specific models.
> > >
> > > ## QS3: Distinguish o1-series models with regular LLM in Related Work
> > >
> > > Thank you for your comment. We have added a detailed description in Section 2.1 to distinguish o1 from regular LLMs, highlighting its unique capabilities and limitations.
> > >
> > > In Section 2.1, we distinguishes vanilla LLMs, like GPT-4, which are general-purpose language models without specific optimization for reasoning tasks, from specialized reasoning-focused models like o1-mini and o1-preview, which are fine-tuned on reasoning datasets and employ long chain-of-thought during inference to enhance their performance on complex reasoning tasks, though they still face limitations in tackling highly complex planning problems.
> > >
> > > ## QS4: Include the Clarification of Metric in Section 4.1
> > >
> > > Thank you for your helpful suggestion. We have added a detailed clarification about the metrics in Section 4.1 and provided additional explanations in the caption to Table 2.
> > >
> > > ## QS5: Control Tokens for iVML and BoN
> > >
> > > Thank you for your insightful question regarding the performance scaling of BoN and the added benefit of iVML.
> > >
> > > As shown in the table, BoN achieves high performance with a relatively small number of tokens. For instance, with 100k tokens, BoN attains an accuracy of **95.8**, which is quite impressive. This indicates that BoN is highly effective when the computational resources are limited and the number of samples is relatively small.
> > >
> > > However, as the number of tokens increases, the performance of BoN starts to plateau. On the other hand, iVML demonstrates a different trend. With 100k tokens, iVML achieves an accuracy of **78.4**, which is lower than BoN. However, as the number of tokens increases, iVML's performance improves significantly. With 500k tokens, iVML's accuracy reaches **90.1**, and with 2M tokens, it further increases to **97.5**. This indicates that iVML has a higher upper limit in performance when more computational resources are available.
> > >
> > > ### NL2Domain Experiment with Controlled Tokens
> > >
> > > | # Tokens | BoN | iVML |
> > > | --- | --- | --- |
> > > | 100k | **95.8** | 78.4 |
> > > | 500k | **92.9** | 90.1 |
> > > | 2M | 93.6 | **97.5** |
> > >
> > > ## References
> > >
> > > [1] Kewei Cheng, Jingfeng Yang, Haoming Jiang, Zhengyang Wang, Binxuan Huang, Ruirui Li, Shiyang Li, Zheng Li, Yifan Gao, Xian Li, et al. Inductive or deductive? rethinking the fundamental reasoning abilities of llms. arXiv preprint arXiv:2408.00114, 2024.
> > >
> > > [2] Karthik Valmeekam, Kaya Stechly, and Subbarao Kambhampati. Llms still can’t plan; can lrms? a preliminary evaluation of openai’s o1 on planbench. arXiv preprint arXiv:2409.13373, 2024.
> > > [3] Kevin Wang, Junbo Li, Neel Pratik Bhatt, Yihan Xi, Qiang Liu, Ufuk Topcu, and Zhangyang Wang. On the planning abilities of openai’s o1 models: Feasibility, optimality, and generalizability. ArXiv, abs/2409.19924, 2024. URL https://api.semanticscholar.org/CorpusID:272988023.

---

> > > > ### Comment · Reviewer_CeE4 · 2025-03-30
> > > >
> > > > Thank you for these additional clarifications and revisions. All of my concerns are addressed and I vote for acceptance.

---

### Decision · Action_Editor_JMjg · 2025-04-03

**Recommendation:** Accept as is

**Comment:**

All reviewers unanimously recommend accept after rebuttal

**Audience:**

Shall be of broad interest given the hot trend of LLM planning

**Claims And Evidence:**

The paper proposes a training-free prompting  framework for LLMs to address complex planning problems by transforming them into PDDL domains, which serves as a world model and can then be solved by classic searching algorithms. The idea is not surprisingly new, but solidly executed